# Generalizability of Experimental Studies

## Abstract

Experimental studies are a cornerstone of Machine Learning (ML) research. A common and often implicit assumption is that the study's results will generalize beyond the study itself, e.g., to new data. That is, repeating the same study under different conditions will likely yield similar results. Existing frameworks to measure generalizability, borrowed from the causal inference literature, cannot capture the complexity of the results and the research questions of an ML study. The problem of measuring generalizability in the more general ML setting is thus still open, also due to the lack of a mathematical formalization of experimental studies. In this paper, we propose such a formalization, use it to develop a framework to quantify generalizability, and propose an instantiation based on rankings and the Maximum Mean Discrepancy. The definition we propose is appropriate to compute the generalizability of experimental results within a bounded range of levels. We show how our framework offers insights into the number of experiments necessary for a generalizable study, and how experimenters can benefit from it. Finally, we release the `genexpy` Python package, which allows for the evaluation of the generalizability of other experimental studies.

## 1 Introduction

Experimental studies are a cornerstone of Machine Learning (ML) research. Due to their importance, the community advocates for high methodological standards when performing, evaluating, and sharing studies (Hothorn et al., 2005; Huppler, 2009; Montgomery, 2017).

The quality of an experimental study depends on multiple independent aspects. First, the experimenter should properly define the *scope* and the *research questions* of the study. Particular attention must be given to the choice of benchmarked methods and experimental conditions (Boulesteix et al., 2015; Bouthillier et al., 2021; Dehghani et al., 2021). Second, the study should contain the necessary documentation to be *reproduced* by independent parties. This aspect has recently drawn much attention due to the so-called reproducibility crisis (Baker, 2016; Gundersen et al., 2023; Peng, 2011; Raff, 2023; 2021). Third, the results of the study should be sensibly analyzed to draw conclusions regarding, for instance, the *significance* of the findings (Benavoli et al., 2017; Corani et al., 2017; Demsar, 2006). Finally, the *generalizability* of a study concerns how well its results are replicated under unseen experimental conditions, such as datasets not considered in the study (National Academies of Science, 2019; Findley et al., 2021; Pineau et al., 2021). Significance and generalizability,[1] although sometimes confused, are two independent aspects of a study (Findley et al., 2021): On the one hand, significant findings may not generalize to other conditions; on the other hand, results might consistently be not significant. Appendix A further discusses their differences.

Generalizability captures how close the results are *between* two different samples of experiments. It is, conceptually, closely related to model replicability. A model is $\rho$-replicable if, given i.i.d. samples from the same data distribution, the trained models are the same with probability $1 - \rho$ (Impagliazzo et al., 2022). An experimental study is generalizable if, when repeated under different experimental conditions, the results are likely similar (National Academies of Science, 2019). A quantifiable notion of generalizability thus requires formalizing of experimental studies, of their results, and of similarity between results.

---

[1]Also known as respectively the internal and external validity of a study.

Significance, instead, captures how strong the findings are *within* the specific sample of experiments performed. Multiple publications have shown how different choices of experimental conditions can lead to very different results (Benavoli et al., 2017; Boulesteix et al., 2017; Bouthillier et al., 2021; Dehghani et al., 2021; Gundersen et al., 2022; Mechelen et al., 2023). Some experimental studies have also reported this phenomenon. Matteucci et al. (2023) highlight how previous studies, conducted under different conditions, report different encoders as significantly better than others. Similarly, Lu et al. (2023) re-evaluated coreset learning methods and found that all of the methods they considered did not beat a naïve baseline.

Quantifying generalizability can help determine the appropriate size of experimental studies. While one dataset is intuitively not enough to draw generalizable conclusions (unless all experiments have the same outcome), $10^6$ datasets likely are. Of course, such large studies are usually not practical: it is crucial to determine the minimum amount of data needed to achieve generalizability. This principle also applies to other experimental factors, such as initialization seed, task, or quality metric.

We propose what, to our knowledge, is the first methodology to quantify the generalizability of an experimental study tailored to the ML setting. The core idea of our approach is to assume the existence of a true distribution of experimental results. Under this assumption, running experiments reduces to sampling from the true distribution. Furthermore, the closer the sample's empirical distribution is to the true distribution, the more generalizable the results are. Specifically, our contributions are the following:

1. A novel measure-theoretic formalization of experimental studies.
2. A quantifiable definition of the generalizability of experimental studies.
3. A closed-form approximation of the Maximum Mean Discrepancy.
4. A workflow to estimate the size of a study to obtain generalizable results.
5. The analysis of two experimental studies, Matteucci et al. (2023); Srivastava et al. (2023).
6. The GENEXPY[2] Python module, a tool for experimenters to effortlessly evaluate their studies.

Paper outline: Section 2 discusses the related work, Section 3 formalizes experimental studies, Section 4 defines generalizability and provides the algorithm to estimate the required size of a study for generalizability, Section 5 contains the case studies, and Section 6 describes the limitations and concludes.

## 2 Related work

We first discuss the literature related to the problem we are tackling, i.e., why experimental studies may not generalize. Second, we overview the existing concept of model replicability, closely related to our work. Finally, we include other meanings that these words can assume in other domains.

**Non-generalizable results.** It is well known that experimental results can significantly vary based on design choices (Lu et al., 2023; Matteucci et al., 2023; Qin et al., 2023; McElfresh et al., 2022). Possible reasons include an insufficient number of datasets (Dehghani et al., 2021; Matteucci et al., 2023; Alvarez et al., 2022; Boulesteix et al., 2015) as well as differences in hyperparameter tuning (Bouthillier et al., 2021; Matteucci et al., 2023), initialization seed (Gundersen et al., 2023), and hardware (Zhuang et al., 2022). As a result, the statistical benchmarking literature advocates for experimenters to motivate their design choices (Bartz-Beielstein et al., 2020; Mechelen et al., 2023; Boulesteix et al., 2017; Bouthillier et al., 2021; Montgomery, 2017) and clearly state the research questions they are attempting to answer with their study (Bartz-Beielstein et al., 2020; Moran et al., 2023).

**Replicability and generalizability in ML.** Our work formalizes and extends the definitions of replicability and generalizability given in Pineau et al. (2021) and National Academies of Science (2019). Intuitively, replicable work consists of repeating an experiment on different data, while generalizable work varies other factors as well—e.g., task, seed. In ML, these terms are usually associated to learning algorithms rather than experimental studies. A generalizable model has small generalization error on unseen data (McElfresh

---

[2]The module is already published on PyPI, which, however, is not anonymized. For this reason, we add the link to an anonymized directory containing all of the code: `https://anonymous.4open.science/r/genexpy-5B3A`

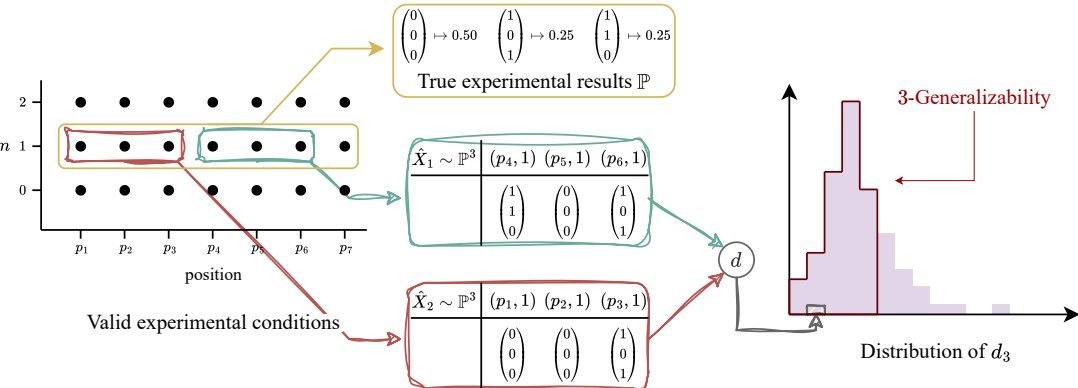

Figure 1: The 3-Generalizability of the "checkmate-in-one" task (see Example 3.1), as the probability for two realizations to yield similar results according to some distance $d$, with $d_3 := d(X, Y)$ if $X, Y \sim \mathbb{P}^3$. Note that the design factor $m$ is fixed, while the generalizability factor *position* varies.

et al., 2022), while a replicable model learns the same parameters from different i.i.d. samples (Impagliazzo et al., 2022). Model replicability is also linked to model stability, differential privacy, generalization error, and global stability (Bun et al., 2023; Chase et al., 2023; Ghazi et al., 2023; Moran et al., 2023; Dixon et al., 2023).

**Other scientific fields.** The terms replicability, reproducibility, and generalizability — we will favor this latter — are sometimes used interchangeably to denote the external validity of a study, i.e., the property of a study to produce the same results when repeated under different experimental conditions. Unlike transportability, generalizability concerns the external validity of results when the experimental conditions are sampled from the same population, for which no systematic changes are expected (Findley et al., 2021). Conceptually, generalizability is to out-of-sample generalization what transportability is to out-of-distribution generalization. A study's generalizability can be evaluated in different ways, according to the scientific domain of reference. In causal inference, its main application being in the social and political sciences (Campbell, 1957), generalizability is often assessed using the sign- and effect-generalization of a treatment (Egami & Hartman, 2023). In practice, it is the amount of times that a significant positive or negative effect is observed when the study is repeated. Similar methodologies appear in other domains, such as psychology (Collaboration, 2015; Klein et al., 2018), economics (Camerer et al., 2016), and social sciences Delios et al. (2022).

These methodologies suffer from important limitations. First, they are limited to numerical, and often univariate, response variables. In the usual setting for ML experiments, instead, one can be interested in non-numerical response variables, such as rankings or multiple comparisons (Demsar, 2006). Second, they rely on performing the same statistical test multiple times, possibly without applying appropriate corrections (Madjarova et al., 2022). Statistical tests themselves have problems, for instance, in interpreting their outcome (Benavoli et al., 2017). Appendix A also shows how, even when results are sampled from the same distribution, significant findings might not generalize due to unavoidable random fluctuations.

## 3 A probabilistic model of experimental studies

The key idea of our formalization is that there exists a true distribution of experimental results, and that running an experiment is essentially the same as sampling from this distribution. To build to this intuition, we first introduce the necessary terminology in Section 3.1 and then move to the measure-theoretic definition in Section 3.2. This latter will serve as the foundation for generalizability.

### 3.1 Fundamentals and notation

In a nutshell, an *experimental study* is a collection of *experiments* comparing the same *alternatives* under different *experimental conditions*. An experimental condition is a tuple of *levels* of *experimental factors*, the parameters defining the experiments. An experimental study aims at answering some *research question*.

*Example* 3.1. (The "checkmate-in-one" task, see Figure 1) An experimenter wants to compare three Large Language Models (LLMs), the *alternatives*, on the "checkmate-in-one" task (Srivastava et al., 2023; Ammanabrolu et al., 2019; 2020; Dambekodi et al., 2020). The assignment is to find the unique checkmating move from a position of pieces on a chessboard: an LLM succeeds if and only if it outputs the correct move. The experimenter considers two *experimental factors*: the number of shots, $m$, and the initial position on the chessboard, $p_l$. The experimenter wants to find if LLM $a_1$ ranks consistently (in the same position) against the other two LLMs when changing the initial position, for a fixed number of shots.

**Alternatives.** An alternative $a \in A$ is an object evaluated in the study, like an LLM in Example 3.1. We call $A$ the finite set of alternatives considered in the study, with cardinality $n_a$.

**Experimental factors.** An experimental factor is *anything* that may affect the result of an experiment. We use $i$ to denote a factor, $C_i$ the (possibly infinite) set of *levels* that $i$ can take, and $I$ the set of all factors. We classify the factors into categories adapted from Montgomery (2017, Chapter 1).

- *Held-constant factors* are presumed not to significantly impact the results, they are hence out of the study's scope and are fixed to a single level; examples are "programming language" or "number of cross-validated folds".
- *Design factors* are expected to significantly impact the results and have a relatively small set of levels; examples are "quality metric" or "number of shots".
- *Stochasticity factors* are included in the study only to "average out" the effect of randomness in either the benchmarked algorithms or the experimental conditions; the results are aggregated w.r.t. these factors before proceeding to the analysis; common examples are "seed" or "cv fold".
- *Generalizability factors* ($I_{\text{gen}}$) have a larger number of levels; the experimenter wants to obtain results that generalize to unseen levels of these factors; examples are "dataset" or "chessboard position".

The same factor may play different roles in different studies, according to the studies' objectives (Montgomery, 2017, Chapter 1). A benchmark such as Matteucci et al. (2023), for instance, considers "dataset" as a generalizability factor, as the results are expected to generalize w.r.t. the choice of dataset. A study focusing on the performance of classifiers on a single dataset will consider "dataset" as a held-constant factor.

**Experimental conditions.** An *experimental condition* $\mathbf{c}$ is a tuple of levels of experimental factors, which lives in the universe of *valid experimental conditions* $C$, $\mathbf{c} = (c_i)_{i \in I} \in C \subseteq \prod_{i \in I} C_i$. In our formalization, it plays the role of the sample space for the experimental results (Section 3.2). We assume that: 1. The levels of the experimental factors are chosen independently, as long as the resulting experimental condition is in $C$; 2. A valid experimental condition univocally identifies an experiment.

**Set of experimental results.** An experiment, i.e., an evaluation of all alternatives under one experimental condition, has as result an element of some set $\mathcal{X}$, the *set of experimental results*. The exact nature of $\mathcal{X}$ is to be decided on a case-by-case basis; for instance, a set of rankings, $\mathcal{R}_{n_a}$, or a set of evaluations, $\mathbb{R}^{n_a}$.

**Experimental study.** An *(experimental) study* is a tuple $S = (A, C, I_{\text{gen}}, \mathcal{X}, Q)$, where $A$ is the set of alternatives being compared, $C$ is the set of valid experimental conditions, $I_{\text{gen}}$ is the set of generalizability factors, $\mathcal{X}$ is the space of results, and $Q$ is the set of research questions.

*Example* 3.1 (Continued). The universe of valid experimental conditions is $C = \{(p_l, m)\}_{l,m}$, where $p_l$ is a valid configuration of pieces on a chessboard and $m$ is the non-negative number of shots. As the goal of the experiment involves ranking the LLMs, the experimenter defines the result of an experiment on $(p_l, m)$ as a

ranking of the three LLMs, according to whether or not they output the checkmating move. For instance, suppose that only $a_1$ and $a_2$ output the correct move: the result is then $(0, 0, 1)$, i.e., $a_1$ and $a_2$ are tied best.

## 3.2 A probabilistic model of experimental results

The core aspect of our probabilistic model is a random variable, the *experimental results*. This is possible with mild assumptions.[3]

**Definition 3.1** (Experimental results)**.** The *experimental results of a study $S$* is a random variable $X^S : C \to \mathcal{X}$, whose distribution we indicate with $\mathbb{P}$.

Defining experimental results in this way has multiple benefits. First, intuitively, $\mathbb{P}$ assigns higher probability to those (sets of) results that are more likely to be observed. Second, the distribution $\mathbb{P}$ serves as the "true" distribution of results, i.e., what one would observe by performing experiments on *all* valid experimental conditions. Third, running an experiment on a subset of experimental conditions is the same as sampling from $\mathbb{P}$ and considering the empirical distribution of the sample, as $C$ serves as the sampling space of $X^S$.

# 4 Generalizability of experimental studies

The currently accepted definition of generalizability is the property of two independent studies with the same research question to yield similar results, see National Academies of Science (2019); Pineau et al. (2021). Although intuitive, this notion is not practically useful as it cannot be assessed objectively. We thus propose the following quantifiable definition of generalizability based on our framework (see Section 3).

**Definition 4.1** (Generalizability)**.** Let $\mathbb{P}$ be the result of a study $S = (A, C, I_{\mathrm{gen}}, \mathcal{X}, Q)$ and let $d$ be a pseudo-distance between probability distributions. The *n-generalizability of $S$* is

$$n\text{-}\mathrm{Gen}\,(S; \varepsilon) \coloneqq F_{d_n}(\varepsilon) = \Pr_{X, Y \overset{\mathrm{iid}}{\sim} \mathbb{P}^n} (d(X, Y) \leq \varepsilon), \tag{1}$$

where $\varepsilon \in \mathbb{R}^+$ is a dissimilarity threshold, $\mathbb{P}^n$ is the product measure of $n$ independent copies of $\mathbb{P}$, $F_{d_n}$ is the cdf of the variable $d(X, Y)$, with $X, Y \sim \mathbb{P}^n$.

Intuitively, the *n*-generalizability is the probability, computed with resampling, that two realizations of $S$ yield "similar" results, as defined by $d$ and $\varepsilon$. In practice, sampling results independently at random from $\mathbb{P}$ corresponds to running experiments on randomly selected levels of the generalizability factors. If a generalizability factor is bounded (categorical or numerical and bounded), sampling a level at random and proceeding with the experiment is enough. If a generalizability factor is unbounded, the experimenter has to specify a bounded range of admissible values to sample from. The computed generalizability is then valid for the region of $\mathcal{C}$ defined by the values of the generalizability factors being in the specified ranges.

We can now define when an experimental study is generalizable.

**Definition 4.2** (Generalizable study)**.** Let $S$, $\mathcal{X}, d$ as in definition 4.1, and let $n \in \mathbb{N}$. $S$ is $(\alpha, \varepsilon, n)$-*generalizable* if $n\text{-}\mathrm{Gen}\,(S; \varepsilon) \geq \alpha$.

Definition 4.1 permits an arbitrary distance $d$: depending on this choice, different formalizations of the research questions are possible. In the rest of this section, we propose a concrete instantiation based on the Maximum Mean Discrepancy (Gretton et al., 2006), rankings, and appropriate kernels. It is however important to emphasize that these choices are not binding: any of the components of definition 4.1 may be replaced by another, more suitable to the study at hand.

## 4.1 Experimental results: Rankings with ties

Experimental results can be formalized in different ways, such as raw performance metrics, time series, or rankings. Among these, rankings are arguably one of the most natural forms:

---

[3]Namely, we assume that $C$ is a probability space and that $\mathcal{X}$ is a separable topological space endowed with its Borel $\sigma$-algebra.

(i) Rankings are already widely used for non-parametric tests such as Friedman, Nemenyi, and Conover-Iman (Demsar, 2006; Conover & Iman, 1982).

(ii) Rankings do not suffer from experimental-condition-fixed effects, such as a dataset being inherently easier to solve than another one. Even though there are multiple ways to deal with these effects, there is no preferred one in the literature. See, for instance, the consensus ranking problem (Matteucci et al., 2023; Nießl et al., 2022).

(iii) Rankings allow the definition of interpretable kernels to formalize different research questions of a study, as we illustrate in Section 4.2.

We define rankings (with ties) in the following way.

**Definition 4.3** (Ranking). A ranking $r$ on $A$ is a transitive and reflexive binary endorelation on $A$. Equivalently, $r$ is a totally ordered partition of $A$ into *tiers* of equivalent alternatives. $r(a)$ denotes the *rank* of $a \in A$, i.e., the position of the tier of $a$ in the ordering. The space of rankings of $n_a$ alternatives is $\mathcal{R}_{n_a}$.

## 4.2 Research questions: Kernels

Research questions act as lenses, focusing on specific aspects of the results that are of interest for the experimenter. For instance, consider the research question in Example 3.1: *"Is $a_1$ consistently in the same position of the ranking?"*. In this case, the research question solely focuses on $a_1$'s position in the rankings, ignoring the positions of $a_2$ and $a_3$. On the contrary, the research question *"Are the best alternatives consistently the same ones?"* focuses only on the top tier of the ranking, ignoring the rest. Changing the research question of a study can thus heavily impact how the results are analyzed, and should be reflected by the distance $d$ being used. The MMD can account for this by choosing an appropriate kernel, a symmetric and positive definite function $\kappa : \mathcal{X} \times \mathcal{X} \to \mathbb{R}^+$. In the following, we describe three kernels for rankings, covering three representative research questions. Additional details, including the definition of the RBF kernel for vector data, are in Appendix B.

**Borda kernel.** The Borda kernel is suitable for research questions in the form *"Is alternative $a^*$ consistently ranked the same?"*. It is based on the Borda count, defined as the number of alternatives (weakly) dominated by a given one (Borda, 1781). We take the difference between the Borda counts of $a^*$ in both rankings.

$$\kappa_b^{a^*,\nu}(r_1, r_2) = e^{-\nu|b_1 - b_2|},$$

where $b_l = |\{a \in A : r_l(a) \geq r_l(a^*)\}|$ is the number of alternatives dominated by $a^*$ in $r_l$ and $\nu \in \mathbb{R}^+$. The Borda kernel takes values in $[e^{-\nu n_a}, 1]$. If $\nu$ is too large compared to $1/|b_1 - b_2|$, the kernel is oversensitive and will heavily penalize even small differences. On the contrary, if $\nu$ is too small, the kernel is undersensitive and will not penalize deviations unless they are very large. As $|b_1 - b_2| \in [0, n_a]$, we recommend $\nu = 1/n_a$.

**Jaccard kernel.** The Jaccard kernel is suitable for research questions in the form *"Are the best alternatives consistently the same ones?"*. As it measures the similarity between sets (Gärtner et al., 2006; Bouchard et al., 2013), we use it to compare the top-$k$ tiers of two rankings.

$$\kappa_j^k(r_1, r_2) = \frac{\left| r_1^{-1}([k]) \cap r_2^{-1}([k]) \right|}{\left| r_1^{-1}([k]) \cup r_2^{-1}([k]) \right|},$$

where $r^{-1}([k]) = \{a \in A : r(a) \leq k\}$ is the set of alternatives with rank lower than or equal to $k$. The Jaccard kernel takes values in $[0, 1]$.

**Mallows kernel.** The Mallows kernel is suitable for research questions in the form *"Are the alternatives ranked consistently?"*. It measures the overall similarity between rankings (Jiao & Vert, 2018; Mania et al., 2018; Mallows, 1957). We adapt the original definition in (Mallows, 1957) for ties,

$$\kappa_m^\nu(r_1, r_2) = e^{-\nu n_d},$$

where $n_d = \sum_{a_1,a_2 \in A} |\text{sign}\left(r_1(a_1) - r_1(a_2)\right) - \text{sign}\left(r_2(a_1) - r_2(a_2)\right)|$ is the number of discordant pairs, $|\cdot|$ is the standard absolute value on $\mathbb{R}$, $\text{sign}(x) = x/|x|$, $\text{sign}(0) = 0$, and $\nu \in \mathbb{R}^+$. If a pair is tied in one ranking but not in the other, one counts it as half a discordant pair. The Mallows kernel takes values in $\left[\exp\left(-2\nu\binom{n_a}{2}\right), 1\right]$. If $\nu$ is too large compared to $1/n_d$, the kernel is oversensitive and it will heavily penalize even small differences. On the contrary, if $\nu$ is too small, the kernel is undersensitive and will not penalize deviations unless they are very large. As $n_d \in \left[0, \binom{n_a}{2}\right]$, we recommend $\nu = 1/\binom{n_a}{2}$.

The following example illustrates the three kernels.

*Example* 4.1. Consider two rankings $\mathbf{r} = (0,0,0)$ and $\mathbf{s} = (0,1,1)$, where $\cdot_j$ is the rank of the $j$-th alternative. In $\mathbf{r}$, all three alternatives are tied best in tier 0, while in $\mathbf{s}$ $a_1$ is the best (in tier 0); $a_2$ and $a_3$ are tied worst (in tier 1). To understand their impact on generalizability, consider a study whose result is a distribution assigning both $\mathbf{r}$ and $\mathbf{s}$ probability $1/2$. For the research question corresponding to the Borda kernel, $\mathbf{r}$ and $\mathbf{s}$ answer the research question consistently as $a_1$ weakly dominates all alternatives in both rankings. For the Jaccard and Mallows research questions, instead, the two rankings are either very different ($\kappa_j^1(r_1, r_2) \approx 0.33$) or slightly different ($\kappa_m^{1/3^2}(r_1, r_2) \approx 0.72$). In this simplified case where all three kernels have very similar range, the $n$-generalizability of the study depends mostly on the value of $\kappa(r_1, r_2)$. Hence, we conclude that, for any fixed $\varepsilon$, the study is perfectly generalizable w.r.t. $\kappa_b$, mildly generalizable w.r.t. $\kappa_m$, and not very generalizable w.r.t. $\kappa_j$.

## 4.3 Distance between experimental results: Maximum Mean Discrepancy

In the previous sections, we have formalized an experimental study, its results, and its research questions. The last open point before applying (1) in practice is a definition of $d$, a pseudo-distance between probability distributions. Such a distance should satisfy the following requirements. First, it should take into consideration the research question of a study. Second, it should handle sparse distributions well: empirical studies are typically very small compared to the number of all possible rankings, which grows super-exponentially in the number of alternatives.[4] Third, it should provide a way to indicate the amount of experiments needed to achieve $n$-generalizable results. A distance satisfying the above requirements is the Maximum Mean Discrepancy (MMD) (Gretton et al., 2006; 2012; Mania et al., 2018), of which we give a simplified definition tailored to our use-case of comparing samples of equal size, see (4.1).

**Definition 4.4** (MMD). Let $\mathcal{X}$ be a set endowed with a kernel $\kappa$, and let $\mathbb{P}$ be a probability distributions on $\mathcal{X}$. Let $X, Y \overset{\text{iid}}{\sim} \mathbb{P}^n$, and let $\mathbf{x}, \mathbf{y}$ be two realizations of $X$ and $Y$ respectively. Then, the MMD between the empirical distributions of $\mathbf{x}$ and $\mathbf{y}$ is

$$\text{MMD}\left(\mathbf{x}, \mathbf{y}\right)^2 := \frac{1}{n^2} \sum_{i,j=1}^{n} \kappa(x_i, x_j) + \frac{1}{n^2} \sum_{i,j=1}^{n} \kappa(y_i, y_j) - \frac{2}{n^2} \sum_{i,j=1}^{n} \kappa(x_i, y_j).$$

In the notation above, we define the random variable $\text{MMD}_n$ as

$$\text{MMD}_n := \text{MMD}(X, Y).$$

## 4.4 An interpretable choice of parameters for generalizability

Finally, there remains the choice of an appropriate $\varepsilon^*$ to use in (1). We propose replacing $\varepsilon^*$ with a condition on the desired minimum expected value of the kernel, in the following way. First, we use the following proposition to relate the support of $\text{MMD}_n$ and the extreme values of the kernel.

**Proposition 4.1.** $\text{MMD}_n \in \left[0, \sqrt{2 \cdot (\kappa_{\sup} - \kappa_{\inf})}\right]$, *where* $\kappa_{\sup} = \sup\limits_{x,y \in \mathcal{X}} \kappa(x,y)$ *and* $\kappa_{\inf} = \inf\limits_{x,y \in \mathcal{X}} \kappa(x,y)$.

We now define $\varepsilon^*$ to be in the form of the equation above, replacing $\kappa_{\inf}$ with another value $f_\kappa(\delta^*)$:

$$\varepsilon^* = \sqrt{2\left(\kappa_{\sup} - \delta'\right)}, \tag{2}$$

---

[4]Fubini or ordered Bell numbers, `https://oeis.org/A000670`.

where, as in the proof of Proposition 4.1 (Appendix B.3), we interpret $\delta'$ as the desired average kernel,

$$\delta' = \frac{1}{n^2} \sum_{i,j=1}^{n} \kappa(x_i, y_j).$$

Finally, we define $\delta' = f_\kappa(\delta^*)$, where both $f_\kappa$ and $\delta^*$'s interpretation depends on the kernel. For the three kernels discussed in Section 4.2, with their recommended parameters:

- *Borda kernel.* $\delta^*$ is $|b_1 - b_2| / n_a$, i.e., the difference between the fraction of dominated alternatives between two rankings; $f_{\kappa_b}(x) = e^{-x}$.
- *Jaccard kernel.* $\delta^*$ is the Jaccard coefficient between the top-$k$ tiers of two rankings; $f_{\kappa_j}(x) = 1 - x$.
- *Mallows kernel.* $\delta^*$ is the fraction of discordant pairs; $f_{\kappa_m}(x) = e^{-x}$.

As a concrete example, achieving $(\alpha^* = 0.90, \delta^* = 0.05)$-generalizable results for the Jaccard kernel means that, w.p. 0.90, the average Jaccard coefficient between two rankings drawn from the results is at least 0.95.

### 4.5   An estimate for the necessary number of experiments

When designing a study, the experimenter has to decide how many experiments to run in order to obtain generalizable results. In other words, they need to choose a (minimum) sample size $n^*$ ensuring that the study is $(\alpha^*, \varepsilon^*, n^*)$-generalizable in the sense of definition 4.2:

$$n^* = \min \left\{ n \in \mathbb{N} : n\text{-Gen}(S; \varepsilon^*) \geq \alpha^* \right\}. \tag{3}$$

*Example* 3.1 (Continued). The experimenter wants to obtain results in which, with probability 0.99, $a_1$ dominates the same number of alternatives up to a difference of 1. This does *not* happen, for instance, if $a_1$ dominates all 3 alternatives in one ranking and just 1 (itself) in another one. They therefore choose the Borda kernel with $\nu = 1/n_a$, $\delta^* = 1/3$, and $\varepsilon^* = \sqrt{2}\sqrt{1 - e^{-0.33}}$ as in (2). *How many experiments are enough?*

Let now $F_{\text{MMD}_n}^{-1}(\alpha^*)$ be the $\alpha^*$-quantile of the variable $\text{MMD}_n$ as defined as in definition 4.4. We have observed in our experiments that there exists a power-law relationship between the quantiles of $\text{MMD}_n$ and $n$. An example of this is shown in the bottom row of Figure 2. The following theorem formalizes this concept.

**Theorem 4.2.** *Let $\mathbb{P}$ be a discrete probability distribution, $\kappa$ a kernel, and $\alpha \in [0.6, 1]$. Then, there exists $\beta = \beta(\kappa, \mathbb{P}, \alpha) \in \mathbb{R}$ such that*

$$\log(n) \approx -2 \log \left( F_{\text{MMD}_n}^{-1}(\alpha^*) \right) + \beta. \tag{4}$$

The proof of the Theorem (see Appendix C) relies on obtaining a closed-form approximation of the upper tail of $F_{\text{MMD}_n}$, under the hypotheses of Theorem 4.2 and reminding that $\text{MMD}_n := \text{MMD}(X, Y)$, where $X, Y$. are independent samples of size $n$ from $\mathbb{P}$. The constant $\beta$, whose explicit expression is in Appendix C.1.5, is a real number which depends on $\kappa$, $\mathbb{P}$, and the desired generalizability $\alpha$. We remark that $\beta$ does not depend on $n$, preserving the power-law relationship.

Theorem 4.2 suggests that one can use a small set of $N$ preliminary experiments to estimate $n^*$, iteratively improving the estimate by adding more experimental conditions from a pre-fixed pool. This reflects the reality of many experimental studies, where the experimenter defines the factors and their possible levels upfront. On this result is based Algorithm 1, whose working is illustrated in Figure 2. Appendix D contains a detailed version of the algorithm.

## 5   Case studies

This section shows how to use generalizability to evaluate the results of an experimental study. To do so, we consider each combination of levels of design factors, a *configuration*, independently from the others. The main bottleneck of the estimation is in bootstrapping the distribution of $\text{MMD}_n$. `genexpy` implements it with an algorithm whose run time is in $\mathcal{O}(n_u^2 n_{\text{rep}})$, where $n_u$ is the number of unique results obtained and

---

**Algorithm 1** A workflow for generalizable studies

---

**Require:** $S, \alpha^*, \delta^*$                                 ▷ Study and generalizability thresholds
**Require:** $N_0$                                      ▷ Number of preliminary experiments

    **procedure** RunGeneralizableStudy($S, \alpha^*, \delta^*, N_0$)
        $(\_, \_, \_, \_, \kappa) \leftarrow S$
        $\varepsilon^* \leftarrow f_\kappa(\delta^*)$                                            ▷ see Section 4.4
        $N \leftarrow N_0$
        $\hat{n}_N^* \leftarrow \infty$
        **while** $N < \hat{n}_N^*$ **do**                                 ▷ Until results are generalizable
            $\hat{\mathbb{P}}_N \leftarrow$ RunExperiments($N$)
            $\overline{n} \leftarrow \lfloor N/2 \rfloor$
            **for** $n = 1 \ldots \overline{n}$ **do**
                $F \leftarrow$ GetMMDcdf($\hat{\mathbb{P}}_N, \kappa, n$)
                $q_n \leftarrow F^{-1}(\alpha^*)$
            **end for**
            $\hat{n}_N^* \leftarrow$ Estimate$n^*((n)_{n=1}^{\overline{n}}, (q_n)_{n=1}^{\overline{n}}, \varepsilon^*)$                       ▷ see (4)
            $N \leftarrow N + N_0$
        **end while**
    **end procedure**

---

$n_{\text{rep}}$) is the number of resamples used to compute the MMD. This, combined with numpy optimizations, gave a total run time including all configurations and all 4 kernels) of 7 minutes for the first case study and 10 for the second, averaging at around 2.5 and 1.75 seconds per configuration-kernel, respectively.

### 5.1 Case Study 1: A benchmark of categorical encoders

We now evaluate the generalizability of a recent study (Matteucci et al., 2023) that analyzes the performance of encoders for categorical data. The performance of an encoder is approximated by the quality of a model trained on the encoded data. The *design factors* are "model", "tuning strategy" for the pipeline, and "quality metric" for the model, while the *generalizability factor* is "dataset".

We impute missing values in the results of the study by assigning the worst rank. We evaluate how well the results of the study generalize w.r.t. the following research questions, identified by their respective kernels:

    $\kappa_b$ Does the one-hot encoder (a popular encoder) rank consistently among its competitors?
    $\kappa_j$ Do some encoders outperform all the others?
    $\kappa_m$ Are the encoders overall ranked in a similar order?
  $\kappa_{RBF}$ Are the raw performances of the encoders consistent?

Figure 3 shows the predicted $n^*$ for different choices of $\alpha^*$ and $\delta^*$, the other one fixed at 0.95 and 0.05 respectively. The variance in the boxes comes from the different configurations in the study: different configurations' results may have different distributions of results. We observe on the left that—as expected— obtaining generalizable results requires more experiments as the desired generalizability $\alpha^*$ increases. We can also see that the variance of the boxes increases with $\alpha^*$, meaning that the choice of the design factors has a larger influence on the achieved generalizability. This is also possibly due to tail effects in the estimation of the MMD. We observe the same when decreasing $\delta^*$, as it corresponds to a stricter similarity condition on the rankings. In the rather extreme cases of $\alpha^* = 0.7$ or $\delta^* = 0.3$, even less than 10 datasets are enough to achieve $(\alpha^*, \delta^*)$-generalizability for the rankings. The RBF kernel, on the other hand, exhibits very low $n^*$-s, hinting that the raw performances of the encoders are very similar on different datasets. We can also identify what configurations of the study require more experiments. Let $\alpha^* = 0.95$ and $\delta^* = 0.05$. Consider the research question $\kappa_j$ (Jaccard kernel, stability of the best alternatives) for two configurations $C_1 = (\texttt{LGBM}, \texttt{no tuning}, \texttt{balanced accuracy})$ and $C_2 = (\texttt{logistic regression}, \texttt{full tuning}, \texttt{F1})$. We

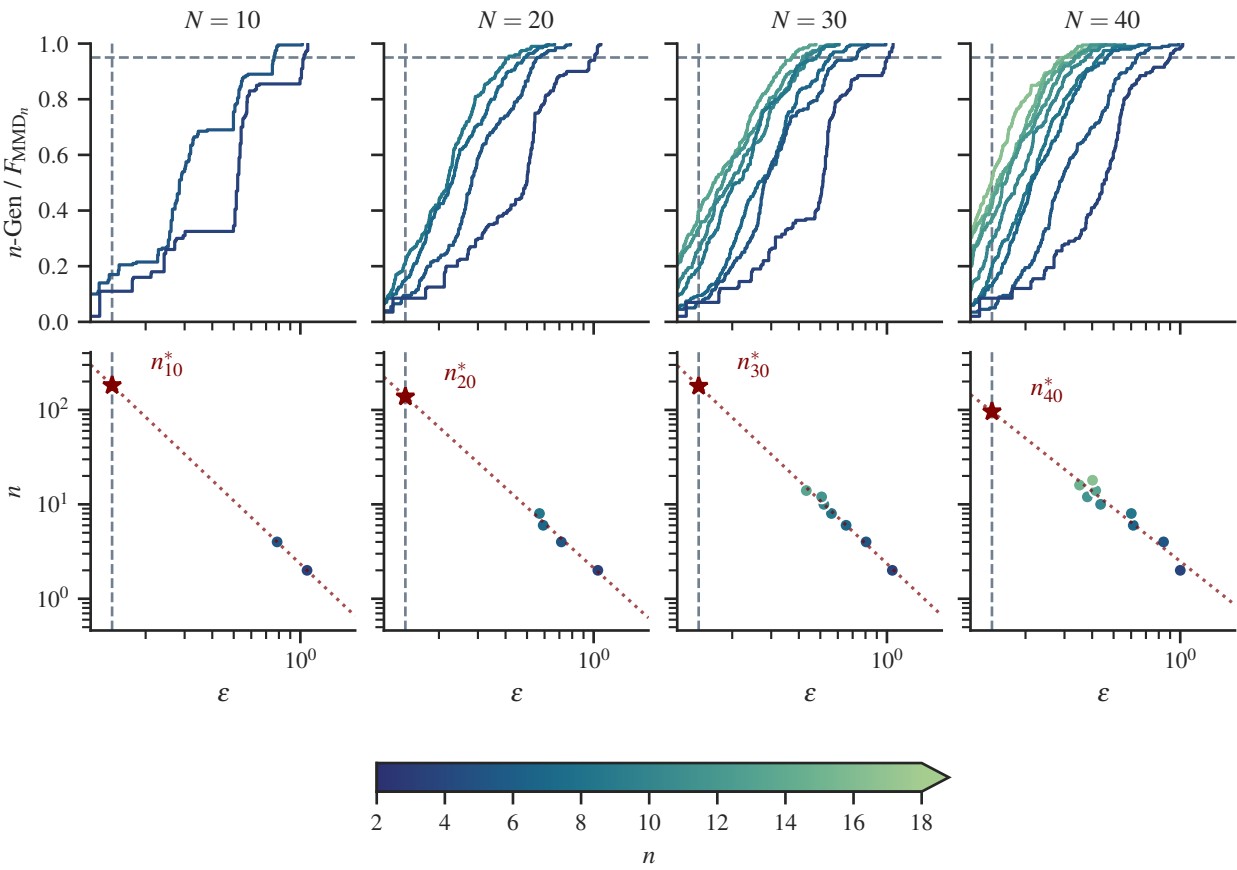

Figure 2: Illustration of Algorithm 1. Top: $n$-generalizability is estimated from $N$ preliminary experiments, for $n \in [2, \ldots, \lfloor N/2 \rfloor]$. Bottom: A power-law relation is fitted to the $\alpha^*$-quantiles of the MMD and $n$, $n_N^*$ is the prediction at $\varepsilon^*$ ($\star$).

estimate that $n^* = 28$ for $C_1$ and $n^* = 47$ for $C_2$. As both configurations were evaluated on 30 datasets, we conclude that the results of $C_1$ are $(\alpha^*, \delta^*)$-generalizable, while those of $C_2$ are not. Hence, the experimenter should prioritize configuration $C_2$ when running additional experiments.

## 5.2 Case study 2: BIG-bench — A benchmark of Large Language Models

We now evaluate the generalizability of BIG-bench (Srivastava et al., 2023), a collaborative benchmark of Large Language Models (LLMs). The benchmark compares LLMs on different tasks, such as the "checkmate-in-one" task (see Example 3.1), and for different numbers of shots. The design factors are "Task" and "number of shots", the generalizability factor is the "subtask" of a task. We stick to the preferred scoring for each subtask. As the results have too many missing values to impute them, we only consider the experimental conditions where at least 80% of the LLMs had results, and to the LLMs whose results cover at least 80% of the conditions.

As before, we evaluate the study's generalizability w.r.t. the following research questions, identified by their respective kernels:

$\kappa_b$  Does GPT3 rank consistently among its competitors?
$\kappa_j$  Do some LLMs outperform all the others?
$\kappa_m$  Are the LLMs overall ranked in a similar order?
$\kappa_{RBF}$  Are the raw performances of the LLMs consistent?

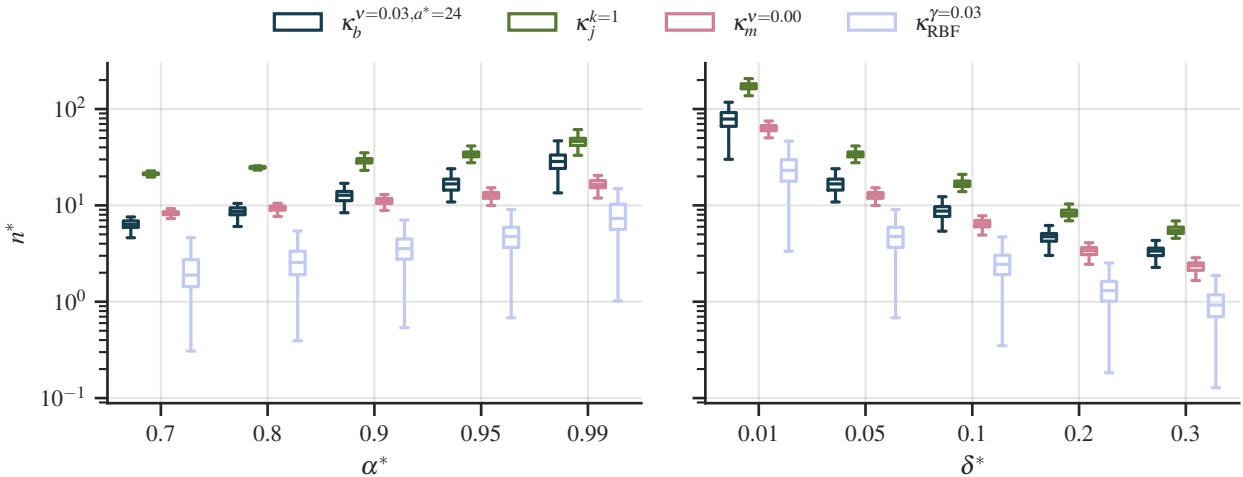

Figure 3: Number of necessary experiments $n^*$ to achieve generalizability for categorical encoders, for different desired generalizability $\alpha^*$, similarity threshold $\delta^*$ (the other fixed at 0.95 and 0.05 resp.), and research questions $\kappa$. The variation in the plot is due to the combinations of design factors.

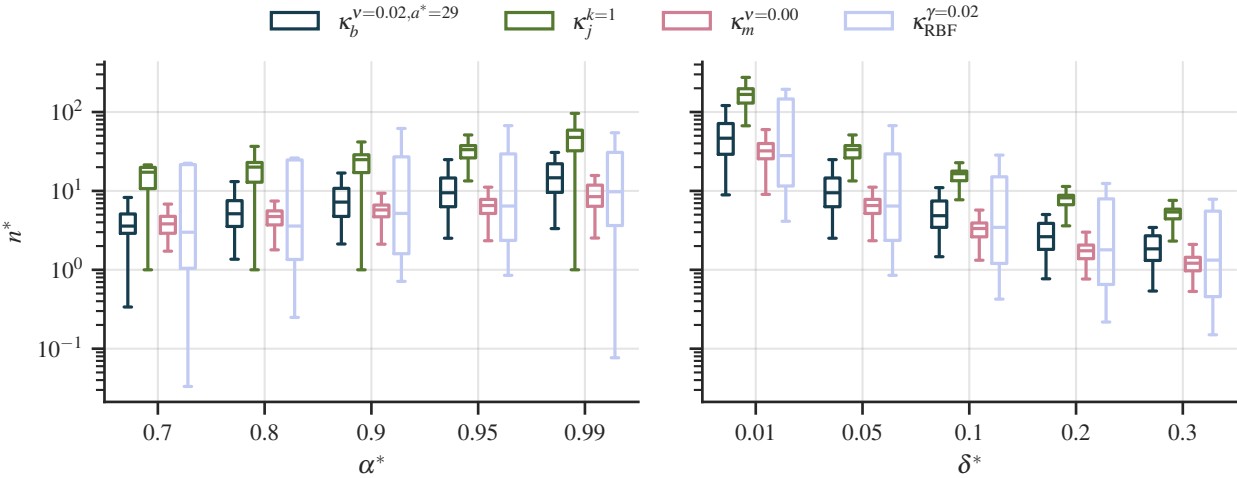

Figure 4: Number of necessary experiments $n^*$ to achieve generalizability for LLMs, for different desired generalizability $\alpha^*$, similarity threshold $\delta^*$ (the other fixed at 0.95 and 0.05 resp.), and research questions $\kappa$. The variation in the plot is due to the combinations of design factors.

Figure 4 shows the predicted $n^*$ for different choices of $\alpha^*$ and $\delta^*$, the other one fixed at 0.95 and 0.05 respectively. Again, the variance in the boxes comes from variance in the design factors, i.e., "task" and "number of shots", and increasing $\alpha^*$ or decreasing $\delta^*$ leads to higher $n^*$. Unlike in the previous section, however, $n^*$ for $\kappa_j$ greatly depends on the configuration, as it can happen that $n^*$ is 1 — or even less, for $\kappa_b$ and $\kappa_{\mathrm{RBF}}$. Consider as above the research question $\kappa_j$ and $(\alpha^*, \delta^*) = (0.95, 0.05)$. We focus on the configurations $C_1 = (\texttt{arithmetic}, 2)$ and $C_2 = (\texttt{conlang\_translation}, 0)$. We estimate $n^* = 65$ for $C_2$. As the study evaluates configuration $C_2$ on 10 subtasks, its results are not generalizable. On the other hand, for $C_1$, we estimate $n^* = 1$: For the research question $\kappa_j$, the 2-shot performance on every subtask of the arithmetic suite is maximized by the same model, PALM-535B. Within our probabilistic framework, this observation implies that the underlying distribution of results, $\mathbb{P}$, places a very high probability on rankings in which PALM-535B occupies the top tier. This interpretation, made on $\mathbb{P}$ rather than on the sample of results available, is in stark contrast to other explanations, such as attributing this pattern to a coincidence. If we

assumed that each of the 44 evaluated LLMs were a priori equally likely to dominate the others on a given sub-task, the probability of observing a single model that is best on all 21 subtasks would be $1/44^{21} \approx 10^{-35}$, an astronomically small value. Consequently, the hypothesis of equal a priori performance is untenable. Instead, our framework treats the empirical results as a sample of the true (but unknown) distribution of experimental results. The estimate derived from the 21 subtasks indicates that PALM-535B is consistently the best model. In other words, given the observed data, the probability that PALM-535B outperforms all other models is close to certainty and the results are $(1, 0)$-generalizable. Finally, our algorithm can retrospectively infer that a single preliminary experiment would have been sufficient to obtain generalizable results. Nevertheless, an estimate of the optimal number of experiments $n^*$ from a single experiment is unreliable. Consequently, the next section examines how the number of preliminary experiments affects the accuracy and stability of the estimated $n^*$.

### 5.3 How many preliminary experiments?

This section investigates how many preliminary experiments are needed to obtain a reliable estimate of the optimal number of trials $n^*$.

We treat the preliminary results as a sample of size $N$ drawn from a known distribution $\mathbb{P}$ over the space of rankings $\mathcal{R}_{n_a}$. The empirical distribution of this sample is denoted $\hat{\mathbb{P}}_N$. Our goal is to assess the accuracy of the procedure described in Section 4.5 for estimating $n^*$ from $N$ independent experiments.

The evaluation pipeline is as follows. First, select a distribution of results $\mathbb{P}$ and requirements $\alpha^* = 0.95$ and $\delta^* = 0.05$. We consider uniform distributions on rankings of $n_a$ alternatives, $U_{n_a}$. Second, calculate $n^*$ using its definition (3). Finally, for a sample size $N \in \{10, 20, 40, 80\}$,

1. Get a sample of size $N$ from $\mathbb{P}$, call its empirical distribution $\hat{\mathbb{P}}_N$.
2. For $n \in \{1, \ldots, \lfloor N/2 \rfloor\}$, bootstrap the distribution of $\mathrm{MMD}_n$ by sampling from $\hat{\mathbb{P}}_N$, repeating this process 100 times.
3. Apply Theorem 4.2 to estimate $\hat{n}_N^*$ for each repetition independently.
4. Compute the ratio $(\hat{n}_N^*)/n^*$ for each repetition independently.

Figure 5 visualizes this ratio: values above 1 indicate over-estimation, below it under-estimation. Results vary with the research question (kernel); for example, estimating $n^*$ for the Borda kernel is noticeably harder and leads to underestimation. Nevertheless, for most repetitions and $N \geq 20$, the empirical estimate $\hat{n}_N^*$ lies within $0.5n^*$ and $2n^*$.

Consequently, our method yields a reliable, order-of-magnitude estimate of $n^*$ from as few as 20 preliminary runs.

## 6 Conclusions

### 6.1 Limitations

First of all, we mainly investigated the instantiation of our framework based on the MMD, rankings, and kernels for rankings. While the type of experimental result is not crucial to our theory, replacing the MMD with another statistical distance might invalidate some properties, mainly Theorem 4.2.

Second, we discuss the key assumptions of the general framework, i.e., of Section 3) and definition 4.1. These assumptions ensure that experiments can be run on levels of the generalizability factors selected at random within a specific region of $\mathcal{C}$, as it is expected when computing generalizability. First, the experimenter should be able to select the levels of all generalizability factors independently from each other. This ensures that, if multiple generalizability factors are involved, it is possible to run experiments on the grid of their levels. Second, as discussed in Section 4, to compute generalizability in practice the experimenter is supposed to run experiments on randomly selected levels of the generalizability factors. If the factors involved are unbounded, such a range must be specified in advance by the experimenter. Any claim of generalizability outside of this range would follow into the scope of transportabiity.

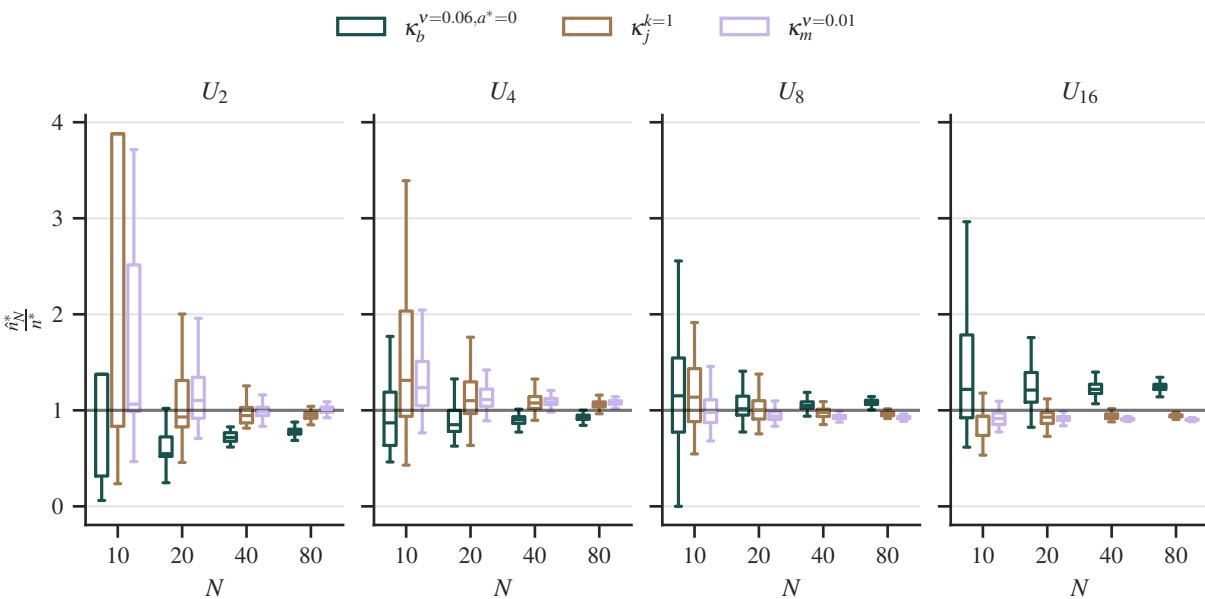

Figure 5: Relative error of the prediction of $n^*$ from $N$ preliminary experiments ($n_N^*$) for uniform distributions of rankings of $n_a$ alternatives $U_{n_a}$.

Other limitations are specific to the MMD-based instantiation, concerning in particular the validity of Theorem 4.2. First, the proof of Theorem 4.2 requires that the experimental results follow the same distribution. This, in turn, requires that the experimental conditions cannot be chosen sequentially, but have to be sampled independently from one another. Theorem 4.2 and the analysis given in Section 5 are therefore likely not to work if the experiments are performed sequentially, for instance in Active Leaning scenarios. Second, from a more practical perspective, the genexpy module in its current implementation can handle only numerical results and rankings. Other data types and kernels require ad-hoc implementations. Third, as discussed in Section 5, the analysis might become unfeasible if the number of unique results observed is too large. This problem can be lessened by "reducing" the set of results, for instance by discretizing it or introducing a minimum performance difference — conceptually similar to the "rope" parameter in Benavoli et al. (2017). Fourth, the proof of Theorem 4.2 (1) requires $\mathbb{P}$ to be discrete and (2) only holds on the upper tail of the distribution of $\mathrm{MMD}_n$. The first assumption is necessary to avoid additional approximations in the first step of the proof (see Appendix C.1.2) and does not play a crucial role in practice, as the observed support of $\mathbb{P}$ is necessarily finite. The second assumption is justified by the fact that experimenters are likely to be more interested in achieving higher levels of generalizability.

To summarize, our framework is designed and suitable to work with a non-sequential factorial experimental design, where the experimenter can run experiments on any combination of levels of generalizability factors, and where the number of generalizability factors is not too large.

## 6.2 Future work

First, incorporating information about the experimental conditions into the framework, for instance, by including information about the datasets. This would allow to study other aspects of external validity, such as transportability, as well as allow for active learning to choose the next experiments to run. Second, based on our experiments in Section 5.3, we intend to provide guarantees and confidence intervals on the convergence of $n_N^*$ to $n^*$. Third, we dealt with missing evaluations by imputing them. Having kernels that can handle missing evaluations might be beneficial. Fourth, rankings, despite their advantages, do not consider the raw performance difference between alternatives. On the other hand, kernels for raw performances (i.e., for vectors in $\mathbb{R}^{n_a}$) lack an obvious interpretation as the goals of a study. Fuzzy rankings may bridge this

gap: performance differences are incorporated into the ranking and the existing kernels for rankings might be adapted to them. Fifth, our framework might help develop tools to identify cherry-picked result. For instance, one can isolate outliers by comparing the results of multiple studies in a meta-analysis fashion. Finally, one can investigate a connection between Probably Approximately Correct (PAC) learning (Valiant, 1984) and generalizability. Namely, as PAC theory makes broad statements valid for all distributions on a given set $\mathcal{X}$ (interpreted as space of results), it might yield lower or upper bounds on the generalizability of any distribution on $\mathcal{X}$, according to whether $\mathcal{X}$ is "very easy" or "very hard". Further optimizations to the computation of the MMD can improve scalability, for instance using Nyström sampling.

### 6.3 Conclusions

An experimental study is generalizable if, with high probability, its findings will hold under different experimental conditions. A formal and general mathematical framework for generalizability is necessary, as non-generalizable studies might be of limited use or even misleading. This paper is, to our knowledge, the first to develop a quantifiable notion for the generalizability of experimental studies. We achieve this by framing experimental studies in probability theory, and we propose a concrete instantiation of our definitions, based on well-known concepts such as kernels and the Maximum Mean Discrepancy. Our approach allows us to estimate the number of experiments needed to achieve a desired level of generalizability in new experimental studies. We demonstrate its utility discussing examples of generalizable and non-generalizable results in two recent experimental studies. Finally, we share a Python module to allow for immediate application of our methodology to other experimental studies.

### Acknowledgments

. . .

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

# A   A toy experiment in significance

Goal of this section is to show how significance and generalizability are different concepts, capturing different aspects of experimental results.[5] In particular, we want to show how significance captures what happens *within* a sample, while generalizability captures aspects of the underlying distribution: how samples of it compare *to one another*. To do so, we explore the following toy scenarios: We are comparing $n_a = 5$ alternatives on a sample of $n = 20$ experimental conditions, where the space of results is the set of permutations of 5 elements, $S_5$,[6] and distribution of true results $\mathbb{P}$, with support $S_5$, is known.

$$\mathbb{P} : (01234) \mapsto 0.55$$
$$(10234) \mapsto 0.45.$$

We set off to answer the following research question: "Is there an alternative that is better than the others?" We repeat the following steps 1000 times:

1. Get a sample from $\mathbb{P}$.
2. Check if the sample is Friedman-significant[7] and Conover-Iman-significant.[8]
3. Compute the 10-generalizability of the sample's empirical distribution.[9]

The results are summarized in Table 1 and Figure 6.

First, we want to show that a sample of results being significant does not imply that other samples are. Even worse, when two samples are significant, they might still disagree on which alternative is the preferred one. The result is unambiguous: all of the samples are Friedman-significant, and 333 of those are also Conover-Iman-significant. Of these, 276 report 0 as the preferred alternative, while the remaining 57 report alternative 1. We stress that both 0 and 1 are the significantly best alternatives in the respective samples, hinting at that significance does not provide information about what is happening in other samples.

Second, we ought to show that the distribution of experimental results can be generalizable even if the results are not significant. Figure 6 (right) depicts how generalizability depends mainly on what alternative is the best, while the effect of significance on it is far weaker.

Table 1: Summary table of the experiment.

| C-I-Significant | Best alternative ($a_{\text{best}}$) | Count | 10-Generalizability |
|---|---|---|---|
| False | 0 | 335 | 0.73 (0.10) |
|  | $\{0, 1\}$ | 150 | 0.82 (0.03) |
|  | 1 | 182 | 0.71 (0.10) |
| True | 0 | 276 | 0.74 (0.11) |
|  | 1 | 57 | 0.72 (0.09) |
| — | — | 1000 | 0.74 (0.10) |

# B   Kernels

## B.1   Kernels for rankings

This section contains the proofs to show that the functions introduced in Section 4.2 are kernels, i.e., symmetric and positive definite. As symmetry is a clear property of all of them, we only discuss their

---

[5]The experimental framework is described in Section 3; generalizability is defined and instantiated as in Section 4.

[6]We indicate an element of $S_5$ with ($abcde$), meaning that alternative $a$ is preferred to $b$ and so on.

[7]$p = 0.05$ for all tests.

[8]Meaning that the best alternative is significantly better than *every* other alternatives.

[9]For the Jaccard kernel (Section 4.2) with $k = 1$, $\delta^* = 0.05$.

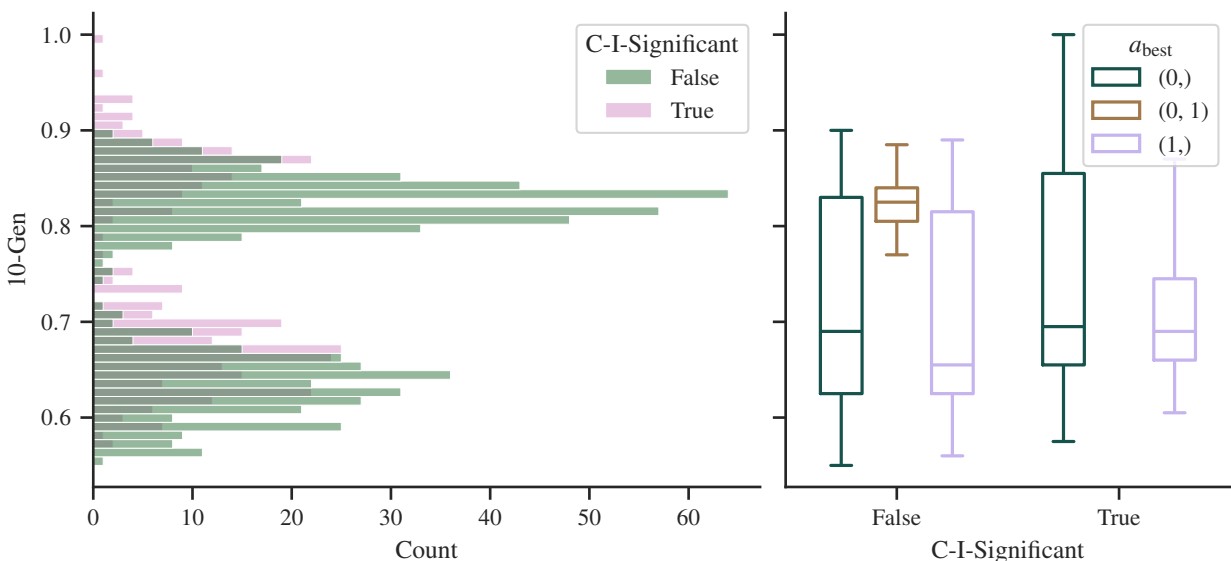

Figure 6: Distribution of 10-generalizability of samples from $\mathbb{P}$, conditioned on: Left: Conover-Iman-significance; Right: Conover-Iman-significance and preferred alternative $a_{\text{best}}$.

positive definiteness. Our proofs for the Borda and Mallows kernels follow Jiao & Vert (2018): we define a distance $d$ on the set of rankings $\mathcal{R}_{n_a}$ and show that $(\mathcal{R}_{n_a}, d)$ is isometric to an $L_2$ space. This ensures that $d$ is a conditionally positive definite (c.p.d.) function and, thus, that $e^{-\nu d}$ is positive definite (Schoenberg, 1938; Schölkopf, 2000). Our proof for the Jaccard kernel, instead, follows without much effort from previous results. For ease of reading, we restate the definitions as well.

**Definition B.1** (Borda kernel).

$$\kappa_b^{a^*,\nu}(r_1, r_2) = e^{-\nu|b_1 - b_2|}, \tag{5}$$

where $b_l = |\{a \in A : r_l(a) \geq r_l(a^*)\}|$ is the number of alternatives dominated by $a^*$ in $r_l$ and $\nu \in \mathbb{R}$.

**Proposition B.1.** *The Borda kernel as defined in* (5) *is a kernel.*

*Proof.* Define a distance

$$d : \mathcal{R}_{n_a} \times \mathcal{R}_{n_a} \to \mathbb{R}^+$$
$$(r_1, r_2) \mapsto \|b_1 - b_2\|,$$

where $b_l = \{a \in A : r_l(a) \geq r_l(a^*)\}$ is the number of alternatives dominated by $a^*$ in $r_l$. Now, $(\mathcal{R}_{n_a}, d)$ is isometric to $(\mathbb{R}, \|\cdot\|_2)$ via the map $r_l \mapsto b_l$. Hence, $d$ is c.p.d. and $\kappa_b$ is a kernel. $\qquad\square$

**Definition B.2** (Jaccard kernel).

$$\kappa_j^k(r_1, r_2) = \frac{\left|r_1^{-1}([k]) \cap r_2^{-1}([k])\right|}{\left|r_1^{-1}([k]) \cup r_2^{-1}([k])\right|}, \tag{6}$$

where $r^{-1}([k]) = \{a \in A : r(a) \leq k\}$ is the set of alternatives whose rank is better than or equal to $k$.

**Proposition B.2.** *The Jaccard kernel as defined in* (6) *is a kernel.*

*Proof.* It is already know that the Jaccard coefficients for sets is a kernel (Gärtner et al., 2006; Bouchard et al., 2013). As the Jaccard kernel for rankings is equivalent to the Jaccard coefficient for the $k$-best tiers of said rankings, the former is also a kernel. $\qquad\square$

**Definition B.3** (Mallows kernel)**.**

$$\kappa_m^\nu(r_1, r_2) = e^{-\nu n_d}, \tag{7}$$

where $n_d = \sum_{a_1, a_2 \in A} |\text{sign}(r_1(a_1) - r_1(a_2)) - \text{sign}(r_2(a_1) - r_2(a_2))|$ is the number of discordant pairs and $\nu \in \mathbb{R}$.

**Proposition B.3.** *The Mallows kernel as defined in* (7) *is a kernel.*

*Proof.* The number of discordant pairs $n_d$ is a distance on $\mathcal{R}_{n_a}$ (Snell & Kemeny, 1962). Consider now the mapping of a ranking into its adjacency matrix,

$$\Phi : \mathcal{R}_{n_a} \to \{0, 1\}^{n_a \times n_a}$$
$$r \mapsto (\mathbf{1}(r(i) \leq r(j)))_{i,j=1}^{n_a},$$

where $\mathbf{1}$ is the indicator function. Then,

$$n_d = \|\Phi(r_1) - \Phi(r_2)\|_1 = \|\Phi(r_1) - \Phi(r_2)\|_2^2$$

where $\|\cdot\|_p$ indicates the entry-wise matrix $p$-norm and the equality holds because the entries of the matrices are either 0 or 1. As a consequence, $(\mathcal{R}_{n_a}, n_d)$ is isometric to $(\mathbb{R}^{n_a \times n_a}, \|\cdot\|_2)$ via $\Phi$. Hence, $n_d$ is c.p.d. and $\kappa_m$ is a kernel. $\qquad\square$

### B.2 Kernels for numeric data

To show how our framework adapts to other spaces of experimental results, i.e., other choices of $\mathcal{X}$, we include the RBF kernel in the case studies, using the scikit-learn implementation (Schölkopf & Smola, 2002; Pedregosa et al., 2011). More specifically, recalling that an experiment evaluates all $n_a$ alternatives, consider the space of results $\mathbb{R}^{n_a}$. The RBF (Gaussian) kernel is then defined as follows.

**Definition B.4** (RBF kernel)**.**

$$\kappa_{\text{RBF}}^\gamma(\mathbf{x}_1, \mathbf{x}_2) = e^{-\gamma \|\mathbf{x}_1 - \mathbf{x}_2\|^2},$$

where $\gamma \in \mathbb{R}$.

### B.3 Proof of Proposition 4.1

**Proposition 4.1.** $\text{MMD}_n \in \left[0, \sqrt{2 \cdot (\kappa_{\text{sup}} - \kappa_{\text{inf}})}\right]$, *where* $\kappa_{\text{sup}} = \sup_{x,y \in \mathcal{X}} \kappa(x, y)$ *and* $\kappa_{\text{inf}} = \inf_{x,y \in \mathcal{X}} \kappa(x, y)$.

*Proof.*

$$0 \leq \text{MMD}(\mathbf{x}, \mathbf{y})^2 = \frac{1}{n^2} \sum_{i,j=1}^n \kappa(x_i, x_j) + \kappa(y_i, y_j) - \frac{2}{n^2} \sum_{i,j=1}^n \kappa(x_i, y_j)$$

$$\leq \frac{2}{n^2} \sum_{i,j=1}^n \kappa_{\text{sup}} - \frac{2}{n^2} \sum_{i,j=1}^n \kappa_{\text{inf}}$$

$$= 2(\kappa_{\text{sup}} - \kappa_{\text{inf}})$$

$$\square$$

## C A power-law relation for the MMD

This Section discusses the power-law relation between $n$ and $F_{MMD_n}^{-1}(\alpha)$, see Theorem 4.2. We will derive the relation in two independent ways. The first, shown in Section C.1, uses a closed-form approximation for the cumulative of $\text{MMD}_n$. There is existing work on the topic, Rustamov (2019), which focuses on (1) numerical distributions, (2) one of which is a multivariate normal, and (3) considering the Gaussian kernel.

Our approximation, instead, relies on the hypotheses of Theorem 4.2, i.e., (1) that the distributions are discrete and (2) we are in the upper tail of the distribution.

The second, shown in Section C.2 discusses a similar law which is a direct consequence of a bound introduced in Gretton et al. (2012). Remarkably, both power laws are in the form

$$\log n = -2 \log F_{\mathrm{MMD}_n}^{-1}(\alpha^*) + \beta,$$

hinting at that the first coefficient $(-2)$ is a fundamental constant for the MMD. We do not investigate this further in this paper.

### C.1 A closed-form approximation for the biased MMD

This Section shows how to obtain a closed-form approximation for the biased MMD V-statistic, using approximations already existing in the literature. To our knowledge, this is the first time that a closed-form approximation of the MMD is proposed.

Let $\mathrm{MMD}_n = \mathrm{MMD}(\mathbf{X}, \mathbf{Y})$ if $\mathbf{X}, \mathbf{Y} \overset{\text{iid}}{\sim} \mathbb{P}^n$ for some distribution $\mathbb{P}$ with support $\mathcal{X}$, with $l = |\mathcal{X}| \in \mathbb{N} \cup \{\infty\}$. Also, let $F_V$ denote the cumulative density function (cdf) of a random variable $V$, and let $F_V^{-1}$ indicate its inverse cumulative density function (icdf; quantile function). In the following subsections, we:

C.1.1 Approximate $F_{\mathrm{MMD}_n}$ with $F_{\sqrt{Q/n}}$, where $Q \sim 2 \sum_{k=1}^{l} \lambda_k Z_k^2$ and $Z_k \overset{\text{iid}}{\sim} \mathcal{N}(0,1)$.
C.1.2 Approximate $F_Q$ with $F_Y$, where $Y \sim a\chi^2(k)$.
C.1.3 Find $F_Y = g^{-1}(F_Z)$ with $F_Z$, where $Z \sim \mathcal{N}(\mu, \sigma^2)$.
C.1.4 Approximate $F_Z^{-1}$ with a closed-form function $\tilde{\Phi}^{-1}$.

This will yield, in Section C.1.5, an approximation in the form

$$F_{\mathrm{MMD}_n}^{-1}(\alpha) \approx \frac{1}{\sqrt{n}} (g^{-1} \circ \tilde{\Phi}^{-1}(\alpha))^{\frac{1}{2}},$$

from which we will derive

$$\log n \approx -2 \log F_{\mathrm{MMD}_n}^{-1}(\alpha) + \beta_0,$$

thus proving Theorem 4.2.

#### C.1.1 Limiting distribution of $\mathrm{MMD}_n$

The first step is to approximate the cdf of $\mathrm{MMD}_n$ by using its limiting properties. As opposed to the unbiased case treated in, for instance, Gretton et al. (2009; 2012), we are looking for the limiting distribution of a V-statistic $V_n = n \mathrm{MMD}_n^2$.

**Theorem C.1** (Limiting distribution of $\mathrm{MMD}_n$). *Let $\mathbb{P}$ be a distribution with support $\mathcal{X}$, $|\mathcal{X}| = l \in \mathbb{N} \cup \{\infty\}$, consider two random vectors $\mathbf{X}, \mathbf{Y} \overset{\text{iid}}{\sim} \mathbb{P}^n$, and the random variable $\mathrm{MMD}_n = \mathrm{MMD}(\mathbf{X}, \mathbf{Y})$. Then*

$$n \mathrm{MMD}_n^2 \overset{d}{\to} 2 \sum_{k=1}^{l} \lambda_k Z_k^2,$$

*where $\lambda_k$ are the eigenvalues of the operator*

$$T : g \mapsto \int_{\mathcal{X}} \tilde{\kappa}(x, \cdot) d\mathbb{P}(x).$$

*Proof.* Before beginning with the proof, we recall that the results in Serfling (1980) Chapter 6 are stated for $\mathcal{X} = \mathbb{R}$. The above limitation is, however, not included in Chapter 5, which deals with the asymptotic theory of U-statistics. As the main result we use, Serfling (1980) Section 6.4.1 Theorem B, ultimately boils down to reducing the V-statistic case to the U-statistic one, we are confident that our results hold for arbitrary $\mathcal{X}$.

Our proof follows Gretton et al. (2012) Theorem 12 and Serfling (1980) Section 6.4.1 Theorem B. First, we recall the following well-known fact (Gretton et al., 2012, eq. (9)). Let $\tilde{\kappa} : x_i, x_j \mapsto \kappa(x_i, x_j) - \mathbb{E}_x[\kappa(x, x_j)] - \mathbb{E}_x[\kappa(x_i, x)] + \mathbb{E}_{x,y}\kappa(x, y)$ be the centered kernel. Then,

$$\text{MMD}^2_\kappa = \text{MMD}^2_{\tilde{\kappa}} = \text{MMD}^2(\mathbf{x}, \mathbf{y}) = \frac{1}{n^2} \sum_{i,j=1}^{n} \tilde{\kappa}(x_i, x_j) + \tilde{\kappa}(y_i, y_j) - 2\tilde{\kappa}(x_i, y_j), \tag{8}$$

We will now consider the limiting properties of each addend of (8) separately.

**First addend.** We want to apply Serfling (1980) Section 6.4.1 Theorem B to the V-statistic $V_n = \frac{1}{n^2} \sum_{i,j=1}^{n} \tilde{\kappa}(x_i, x_j)$. In particular, using Serfling's notation, we have:

(i) A functional $T(\mathbb{P}) = \mathbb{E}_{x,y\sim\mathbb{P}}\tilde{\kappa}(x, y) \equiv 0$, with estimator $T(\hat{\mathbb{P}}_n) = V_n$.
(ii) A symmetric kernel $h = \tilde{\kappa}$.
(iii) A remainder term $R_n = T(\hat{\mathbb{P}}_n) - T(\mathbb{P}) - V_n \equiv 0$

Condition $A_2$ (Serfling, 1980, p. 225) is satisfied:

(i) $\text{var}_x(\tilde{\kappa}(x, \cdot)) = 0$ for the properties of the centered kernel.
(i)' $\text{var}_{x,y}(\tilde{\kappa}(x, y)) > 0$ if the kernel is not constant.
(ii) $nR_n \equiv 0$.

The hypotheses of finiteness of the kernel are trivially satisfied if we assume that $\mathcal{X}$ is finite. Then, by Theorem B,

$$nV_n \xrightarrow{d} \sum_{k=1}^{\infty} \lambda_k a_k^2, \tag{9}$$

where $a_k \overset{\text{iid}}{\sim} \mathcal{N}(0, 1)$.

**Second addend.** A similar reasoning shows that the second term converges to

$$\sum_{k=1}^{\infty} \lambda_k b_k^2,$$

where $b_k \overset{\text{iid}}{\sim} \mathcal{N}(0, 1)$ and are independent of the $a_k$.

**Third addend.** Gretton et al. (2012) eq. (28) shows that

$$\frac{1}{n} \sum_{ij=1}^{n} \tilde{\kappa}(x_i, y_j) \xrightarrow{d} \sum_{k=1}^{\infty} \lambda_k a_k b_k, \tag{10}$$

**Reassembling** (8). Recombining (9) and (10) and using basic properties of normal and chi-squares, we obtain:

$$n\,\text{MMD}^2_n \xrightarrow{d} \sum_{k=1}^{\infty} \lambda_k(a_k^2 b_k^2 - 2a_k b_k) \sim 2\sum_{k=1}^{\infty} \lambda_k Z_k^2,$$

where $Z_k \overset{\text{iid}}{\sim} \mathcal{N}(0, 1)$, thus concluding the proof. $\qquad\square$

In the discrete case $l < \infty$, for instance if $\mathcal{X} = \mathcal{R}_{n_a}$ is the set of rankings of $n_a$ alternatives, Theorem C.1 takes the following form.

**Corollary C.2.** *Let $\mathbb{P}$ be a distribution with support $\mathcal{X}$, with $|\mathcal{X}| = l \in \mathbb{N}$ and $\mathbf{p} = (\mathbb{P}(x_k))_{k=1}^{l}$. Consider two random vectors $\mathbf{X}, \mathbf{Y} \overset{\text{iid}}{\sim} \mathbb{P}^n$ and the random variable $\mathrm{MMD}_n = \mathrm{MMD}(\mathbf{X}, \mathbf{Y})$. Finally, let $K = (\kappa(x_i, x_j))_{i,j=1}^{l}$ the kernel matrix, $H_l = I_l - \frac{1}{l}1_{ll}$ the centering matrix, and $\tilde{K} = H_l K H_l$ the centered kernel matrix. Then*

$$n\,\mathrm{MMD}_n^2 \overset{d}{\to} 2\sum_{k=1}^{l} \lambda_k Z_k^2, \tag{11}$$

*where $\lambda_k$ are the eigenvalues of the matrix $\tilde{K} \cdot \mathrm{diag}(\mathbf{p})$.*

*Proof.* The statement follows directly from the previous theorem, with some care in translating the infinite-dimensional operator $T$ to the finite-dimensional $\tilde{K} \cdot \mathrm{diag}(\mathbf{p})$. $\qquad\square$

A consequence of the previous corollary is that we can easily calculate the eigenvalues of the operator, which is not trivial in the general case.

**Corollary C.3** (Approximation of $\mathrm{MMD}_n$)**.** *We can use* (11) *to obtain*

$$\mathrm{MMD}_n \sim Q_n = \sqrt{\frac{Q}{n}},$$

*where $Q \sim 2\sum_{k=1}^{l} \lambda_k Z_k^2$ and $Z_k \overset{\text{iid}}{\sim} \mathcal{N}(0,1)$.*

### C.1.2 Moment-matching $Q$

We now approximate $Q$ with a variable $Y \sim a\chi^2(d)$ for appropriate $a$ and $d$ Solomon & Stephens (1977); Bodenham & Adams (2016), matching the first two noncentral moments of $Q$ and $Y$. It all descends from the following formula for the $m$-th moment of a chi-squared variable with $d$ degrees of freedom (Simon, 2002):

$$\mathbb{E}\left[a^m (\chi^2(d))^m\right] = a^m 2^{dm} \frac{\Gamma(m + \frac{d}{2})}{\Gamma(\frac{d}{2})}. \tag{12}$$

**Moments of $Q$.** Using (12), the independence of the $Z_k$, and properties of the $\Gamma$ function, we obtain

$$\mathbb{E}[Q] = 2\sum_{k=1}^{l} \lambda_k \mathbb{E}[Z_k^2] = 2\sum_{k=1}^{l} \lambda_k$$

$$
\begin{aligned}
\mathbb{E}[Q^2] &= \mathbb{E}\left[4\left(\sum_{k=1}^{l} \lambda_k Z_k^2\right)^2\right] \\
&= 4\sum_{k=1}^{l} \lambda_k^2 \mathbb{E}[(Z_k^2)^2] + 4\sum_{i\neq j=1}^{l} \lambda_i \lambda_j \mathbb{E}[Z_i^2]\mathbb{E}[Z_j^2] \\
&= 4\sum_{k=1}^{l} 4\lambda_k^2 \frac{\Gamma(2+\frac{1}{2})}{\Gamma(\frac{1}{2})} + 4\sum_{i\neq j=1}^{l} \lambda_i \lambda_j \\
&= 12\sum_{k=1}^{l} \lambda_k^2 + 4\sum_{i\neq j=1}^{l} \lambda_i \lambda_j
\end{aligned}
$$

**Moments of $Y$.** Using again (12), we get

$$\mathbb{E}[Y] = ad$$
$$\mathbb{E}[Y^2] = a^2 d(d+2)$$

**System of equations.** Equating the two moments, we solve the following system for $a$ and $d$.

$$\begin{cases} ad = 2 \sum_{k=1}^{l} \lambda_k \\ a^2 d(d+2) = 12 \sum_{k=1}^{l} \lambda_k^2 + 4 \sum_{i \neq j=1}^{l} \lambda_i \lambda_j \end{cases}$$

$$\Downarrow$$

$$\begin{cases} a = \frac{\Lambda_2 - \Lambda_1^2}{\Lambda_1} \\ d = \frac{2\Lambda_1^2}{\Lambda_2 - \Lambda_1^2}, \end{cases}$$

where $\Lambda_1 = \sum_k \lambda_k$ and $\Lambda_2 = 3 \sum_k \lambda_k^2 + \sum_{i \neq j} \lambda_i \lambda_j$.

### C.1.3 The Wilson-Hilferty normalization of $Y$

We now proceed to approximate $Y \sim a\chi^2(d)$ with a normal random variable. To do so, we use the Wilson-Hilferty normalization formula Wilson & Hilferty (1931).

$$g(Y) \approx \frac{\left(\frac{Y}{ad}\right)^{\frac{1}{3}} - \left(1 - \frac{2}{9d}\right)}{\left(\frac{2}{9d}\right)^{\frac{1}{2}}}$$

and its inverse (assuming $a, d$ known)

$$g^{-1}(Z) \approx ad \left( \left(\frac{2}{9d}\right)^{\frac{1}{2}} Z + \left(1 - \frac{2}{9d}\right) \right)^3.$$

### C.1.4 Approximating $F_Z$

There exists a wide body of literature on approximations of the cdf of a standard normal variable, $F_Z$. Refer, for instance, to Choudhury et al. (2007). In our work, we settled on Lin (1989), mainly for two reasons: 1. It provides a good approximation in the upper tail ($\alpha \gtrsim 0.5$); 2. It is easily invertible. We denote with $\tilde{\Phi}$ the approximated cdf.

The approximation is as follows:

$$F_Z(x) \approx \tilde{\Phi}(x) = 1 - 0.5e^{-0.717x - 0.416x^2}, x > 0.$$

Inverting it with Mathematica (Inc., 2025) to get the icdf:

$$\tilde{\Phi}^{-1}(\alpha) = c_0 + c_1 \sqrt{c_2 - c_3 \log 2(1 - \alpha)}, \tag{13}$$

where $c_0 = -0.861779, c_1 = 0.00120192, c_2 = 514089, c_3 = 1664000$.

### C.1.5   A closed-form approximation for the icdf of $\mathrm{MMD}_n$

In the notation introduced in the above subsections, we obtain

$$
\begin{aligned}
\log F_{\mathrm{MMD}_n}^{-1}(\alpha) &\approx -\frac{1}{2}\log n + \frac{1}{2}\log(g^{-1}\circ\tilde{\Phi}^{-1}(\alpha)) \\
&= -\frac{1}{2}\log n + \frac{1}{2}\left(3\log C + \log(ad)\right),
\end{aligned} \tag{14}
$$

where

- $C = \left(\sqrt{\frac{2}{9d}}\left(c_0 + c_1\sqrt{c2\log 2(1-\alpha)}\right) + \left(1 - \frac{2}{9d}\right)\right)^3$,

- $c_0, ..., c_4$ are as in (13),

- $a = \frac{\Lambda_2 - \Lambda_1^2}{\Lambda_1}$,

- $d = \frac{2\Lambda_1^2}{\Lambda_2 - \Lambda_1^2}$,

- $\Lambda_1 = \sum_k \lambda_k$, and $\Lambda_2 = 3\sum_k \lambda_k^2 + \sum_{i\neq j}\lambda_i\lambda_j$.

Finally, we invert (14) to find

$$
\log n \approx -2\log F_{\mathrm{MMD}_n}^{-1}(\alpha^*) + 3\log C + \log(ad).
$$

### C.2   Distribution-free power law

Finally, we discuss a power law for $\mathrm{MMD}_n$ which is independent of the distribution $\mathbb{P}$. Although exact, however, this relation offers little practical utility, as using $\bar{\varepsilon}$ as a replacement for $F_{\mathrm{MMD}_n}^{-1}(\alpha^*)$ in (4) yields to overconservative estimates of $n^\star$.

**Proposition C.4.** *Let $\kappa$ be a kernel, and $\alpha \in [0,1]$. Then, there exist $\bar{\varepsilon} = \bar{\varepsilon}(\kappa,\alpha,n)$ and $\beta = \beta(\kappa,\alpha) \in \mathbb{R}$ such that, for any distribution $\mathbb{P}$,*

*(i) $\bar{\varepsilon} \geq F_{\mathrm{MMD}_n}^{-1}(\alpha^*)$,*
*(ii) $\log n = -2\log\bar{\varepsilon} + \beta$.*

*Proof.* First, we find $\bar{\varepsilon}$ following Gretton et al. (2012, Theorem 8). Let $\mathrm{MMD}_n = \mathrm{MMD}(X,Y)$ with $X,Y \overset{\text{iid}}{\sim} \mathbb{P}^n$, and let $\varepsilon = F_{\mathrm{MMD}_n}^{-1}(\alpha^*)$.

$$
1 - F_{\mathrm{MMD}_n}\left(\varepsilon + \sqrt{\frac{2\kappa_{\sup}}{n}}\right) < \exp\left(-\frac{n}{4\kappa_{\sup}}\varepsilon^2\right)
$$

$$
\Updownarrow (1)
$$

$$
1 - F_{\mathrm{MMD}_n}(\bar{\varepsilon}) < \exp\left(-\frac{4\kappa_{\sup}}{n}\left(\bar{\varepsilon} - \sqrt{\frac{2\kappa_{\sup}}{n}}\right)^2\right)
$$

$$
\Updownarrow (2)
$$

$$
F_{\mathrm{MMD}_n}\left(n^{-\frac{1}{2}}\left(\sqrt{-4\kappa_{\sup}\log(1-\alpha)} + \sqrt{2\kappa_{\sup}}\right)\right) \geq \alpha,
$$

where:

(1) $\overline{\varepsilon} := \varepsilon + \sqrt{2\kappa_{\text{sup}}/n}$.

(2) $\alpha := 1 - \exp\left(-\frac{n}{4\kappa_{\text{sup}}}\left(\overline{\varepsilon} - \sqrt{\frac{2\kappa_{\text{sup}}}{n}}\right)^2\right) \Rightarrow \overline{\varepsilon} = n^{-\frac{1}{2}}\left(\sqrt{-4\kappa_{\text{sup}}\log(1-\alpha)} + \sqrt{2\kappa_{\text{sup}}}\right)$.

Thus, taking the logarithms,

$$\log n = -2\log\overline{\varepsilon} + 2\log\left(\sqrt{-4\kappa_{\text{sup}}\log(1-\alpha)} + \sqrt{2\kappa_{\text{sup}}}\right).$$

$\square$

## D   A workflow for generalizable studies

This Section contains Algorithm 2, the extend version of Algorithm 1.

---

**Algorithm 2** A workflow for generalizable studies

---

**Require:** $S, \alpha^*, \delta^*$            ▷ Study and generalizability thresholds
**Require:** $N_0$            ▷ Number of preliminary experiments

    **procedure** RUNGENERALIZABLESTUDY$(S, \alpha^*, \delta^*, N_0)$
        $(\_, \_, \_, \_, \kappa) \leftarrow S$
        $\varepsilon^* \leftarrow f_\kappa(\delta^*)$          ▷ see Section 4.4
        $N \leftarrow N_0$
        $\hat{n}_N^* \leftarrow \infty$
        **while** $N < \hat{n}_N^*$ **do**          ▷ Until results are generalizable
            $\hat{\mathbb{P}}_N \leftarrow$ RUNEXPERIMENTS$(N)$
            $\overline{n} \leftarrow \lfloor N/2 \rfloor$
            **for** $n = 1 \ldots \overline{n}$ **do**
                $F \leftarrow$ GETMMDCDF$(\hat{\mathbb{P}}_N, \kappa, n)$
                $q_n \leftarrow F^{-1}(\alpha^*)$
            **end for**
            $\hat{n}_N^* \leftarrow$ ESTIMATE$n^*((n)_{n=1}^{\overline{n}}, (q_n)_{n=1}^{\overline{n}}, \kappa, \delta^*)$      ▷ see Section 4.5
            $N \leftarrow N + N_0$
        **end while**
    **end procedure**

    **procedure** RUNEXPERIMENTS$(N)$
        **return** $\hat{\mathbb{P}}_N$     ▷ Empirical distribution of the experimental results on $N$ experimental conditions.
    **end procedure**

    **procedure** GETMMDCDF$(\mathbb{P}, \kappa, n)$
        $m \leftarrow$ empty list
        **for** $n = 1 \ldots n_{\text{rep}}$ **do**
            Sample without replacement $(x_j)_{j=1}^{2n} \sim \mathbb{P}$
            $\mathbf{x} \leftarrow (x_j)_{j=1}^{n}$          ▷ Split into disjoint samples
            $\mathbf{y} \leftarrow (x_j)_{j=n+1}^{2n}$
            Append $\text{MMD}_\kappa(\mathbf{x}, \mathbf{y})$ to $m$
        **end for**
        **return** Cumulative Density Function of $m$
    **end procedure**

    **procedure** ESTIMATE$n^*((n)_{n=1}^{\overline{n}}, (q_n)_{n=1}^{\overline{n}}, \varepsilon^*)$
        $\beta_0, \beta_1 \leftarrow$ FITLINEARREGRESSION$((\log n)_{n=1}^{\overline{n}}, (\log q_n)_{n=1}^{\overline{n}})$
        $\hat{n}_N^* \leftarrow \beta_1 \log(\varepsilon^*) + \beta_0$          ▷ see Theorem 4.2
        **return** $\hat{n}_N^*$
    **end procedure**

---

