# OpenReview forum: "Generalizability of Experimental Studies"
_TMLR — Rejected by TMLR_

### Review · Reviewer_ZRGN · 2026-03-20

**Summary Of Contributions:**

The submission provides a mathematical formalisation for measuring the
generalisability of experiments with the focus on experimental studies
in machine learning (ML). The argument is that existing methods to
measure generalisability are not adequate and that the hitherto lack
of a (suitable) mathematical formalisation has prevented
generalizability from being properly measured. Concretely, the authors
claim they can "provide insights" into how many experiments are
required for a generalisable study. Concerning related work, the
authors state that this submission "formalizes and extends the
definitions of replicability and generalizability given in Pineau et
al. (2021) and National Academies of Science (2019)."

**Audience:**

No

**Audience Explanation:**

See previous text box.

**Broader Impact Concerns:**

None.

**Claims And Evidence:**

No

**Claims Explanation:**

On the positive side, what constitutes good practice in empirical
studies is a question worth addressing. The 'reproducibility crisis'
alone is motivation enough. Choosing a particular definition of
'generalizability' and building a formalization around it could
potentially be useful.

The authors state that: "The key idea of our formalization is that
there exists a true distribution of experimental results, and that
running an experiment is essentially the same as sampling from this
distribution". This is how I (and I suspect most people in machine
learning) would immediately conceptualise experimental
variability. So, of course, I have no disagreement with this approach,
but there's nothing new about it. I see no need to "build intuition"
for it. One should generalise to viewing experimental results as
having been sampled on a distribution conditional on the particular
experimental conditions since as the authors state (citing a NAS study)
"An experimental study is generalizable if, when repeated under
different experimental conditions, the results are likely
similar".

So the key idea is uncontroversial and the key definition of
Generalizability (Definition 4.1) is consistent with it. But to make a
substantial and significant contribution to this area a more thorough
connection with existing work is required. There is an ocean of
literature on experimental design dating back, at least, to Fisher's
1935 book. And connecting experimental variability to probability
distributions is standard. The issues which we face when empirically
testing ML methods are not so different from those faced e.g. in
natural science (I doubt the authors would disagree with that point).
Given this, creating a new formalisation and making the claim: "This
paper is, to our knowledge, the first to develop a quantifiable notion
for the generalizability of experimental studies." is unacceptable.

The choice of pseudo-distance between distributions (experimental
results) is key to computing the n-generalizability (a
probability). As the authors note, their particular choices (making use
of MMD, rankings and kernels) are not "binding". I did not find the
arguments for these particular choices compelling.

**Requested Changes:**

I can't see any changes to this paper making it acceptable.

---

> ### Author Response · Authors · 2026-04-03
>
> We thank the reviewer for their insightful comments and suggestions on better locating our work into the existing literature, and we sincerely apologize for the delayed response.
> Following their advice, we downsized our claims of novelty to the ML setting and expanded on the related work.
> The relevant changes are highlighted in orange in the revised manuscript.
> To further improve the quality of our work, we would greatly appreciate more detailed references on what existing work invalidates the novelty of our framework.
>
> ---
>
> *On the positive side, what constitutes good practice in empirical studies is a question worth addressing. The 'reproducibility crisis' alone is motivation enough. Choosing a particular definition of 'generalizability' and building a formalization around it could potentially be useful.*
>
> We thank the reviewer for the kind words.
>
> ---
>
> *The authors state that: "The key idea of our formalization is that there exists a true distribution of experimental results, and that running an experiment is essentially the same as sampling from this distribution". This is how I (and I suspect most people in machine learning) would immediately conceptualise experimental variability. So, of course, I have no disagreement with this approach, but there's nothing new about it. I see no need to "build intuition" for it.".*
>
> We agree that this is a natural formulation and that it is customary to model some aspects of the experimental results as statistics, of which the distributions are then used to perform tests.
> However, as discussed in sections 2, 3, and 4, this is fundamentally different from dealing with distributions on arbitrary spaces.
> Our approach allows for the analysis of a wide range of result into a unified framework, of which we propose an instantiation.
> Moreover, as discussed in section 6, generalizability is but one of its potentialities, which include, for instance, the analysis of transportability.
>
> ---
>
> *One should generalise to viewing experimental results as having been sampled on a distribution conditional on the particular experimental conditions since as the authors state (citing a NAS study) "An experimental study is generalizable if, when repeated under different experimental conditions, the results are likely similar*
>
> We agree with the reviewer that modeling experimental results as conditional probability distributions can be beneficial.
> In the present manuscript we do something that is very closely related: Definition 3.1 defines the result of an experimental study as a random variable defined on a set of valid experimental conditions $\mathcal C$.
> A natural corollary is that one can define a ``substudy'' performed on a $\mathcal D$ of $\mathcal C$ and consider the distribution of its experimental results $Y$.
> Then, one can compare experimental results from different substudies as we discuss in sections 4 and 5.
> While we agree that this is not a theory of conditional distributions, our approach is still natively able to deal with differences in the experimental results from different regions of $\mathcal C$.
>
> Moreover, generalizability, as opposed to transportability, deals with experiments on experimental conditions that are meant to be ``equivalent''.
> With this assumption, we argue that it is natural to assume that the results are also sampled form the same distribution, leaving other analyses (for instance, of how many different distributions of results there are) to future work.
> We discuss this in sections 2 and 6 of the revised manuscript.
>
> ---
>
> *So the key idea is uncontroversial and the key definition of Generalizability (Definition 4.1) is consistent with it. But to make a substantial and significant contribution to this area a more thorough connection with existing work is required. [...]*
>
> We agree that there is plenty of literature on experimental design and, more relevant to the case at hand, result analysis.
> In the light of the reviewer's comment, we updated section 2 and included more references to better locate our work within the existing literature, and we downsized our claim of novelty to the ML scenario.
>
>
> ---
>
> *The choice of pseudo-distance between distributions (experimental results) is key to computing the n-generalizability (a probability). As the authors note, their particular choices (making use of MMD, rankings and kernels) are not "binding". I did not find the arguments for these particular choices compelling.*
>
> As the reviewer correctly points out, these choices serve to illustrate our theoretical framework on actual data, and are mainly suggestions: as mentioned in section 6 of the submitted manuscript, we agree that other choices might be beneficial and we plan on exploring them.
> Finally, understanding why the motivations we bring forward in Section 4 are not compelling would greatly help us improve the quality of our manuscript.

---

### Review · Reviewer_qQqi · 2026-03-24

**Summary Of Contributions:**

This paper proposes a mathematical formalization of ML experimental studies and builds upon it a framework for quantifying generalisability. The authors introduce a concrete instantiation of this framework using ranking metrics and the Maximum Mean Discrepancy. The framework is shown to provide practical guidance on how many experiments are needed to achieve generalizable results, given the assumptions and definitions within the framework. Another note worthy contribution of this paper is  an open-source Python package called genexpy to enable other researchers to apply the introduced framework to evaluate the generalisability of their own studies.

**Additional Comments:**

NA

**Audience:**

Yes

**Audience Explanation:**

I am sure at least some individuals in TMLR's audience would be interested in knowing the findings of this paper, due to it covering such an important topic fundamental to the field. The approach of the this paper in ensuring Experimental Studies are Generalizable is of high importance particularly in the modern ML/AI research climate, with such a high number of papers and little concrete methods to determine  the significance of presented results.

**Claims And Evidence:**

Yes

**Claims Explanation:**

To the best of my knowledge yes. However I do think the discussion around the limitations and assumptions implicit in the framework are very under discussed currently. My recommendation is contingent of a lot more discussion be included here. See requested changes for more details.

**Requested Changes:**

Casual =/= causal, please double check you are using the correct word. This is incorrect in the abstract.

This paper really needs a much longer section talking about its limitations. There are many assumptions made that are not discussed at all in this section. Discussing these assumptions is a keep part of making this paper usable/useful. All the nice formalism and theory means little if people aren't convinced it applies to their experiments. Being more explicit a few larger assumptions I notice with limited discussion at all are (I suspect other reviewers may have noticed others):

1. "The levels of the experimental factors are chosen independently, as long as the resulting experimental condition is in
C"

2. "The proof of the theorem relies on obtaining a closed-form approximation of FMMDn , which we obtain by
reducing MMDn to a normal random variable"

3. There is very limited discussion around settings where you do not think your framework would be appropriate.

I would expect a longish paragraph on each of these.

In figures 3 &4 you have plenty of room for labeling the axis with text rather than symbols is there a reason you choose not to do this?

---

> ### Author Response · Authors · 2026-03-30
>
> We thank the reviewer for their constructive comments and for allowing us to improve the paper's usability. The relevant changes are highlighted in green in the revised manuscript.
>
> ## Limitations
>
> To address their concerns regarding the limitations of our framework, we greatly expanded this section in the revised manuscript.
> In particular, we discuss limitations pertinent to the framework itself (independence and ranges of the generalizability factors) as well as related to the MMD-based instantiation of Section 4 (independence of sampling of experimental conditions, implementation of new results types, scalability, and hypotheses of Theorem 4.2). We also added a brief discussion underneath Theorem 4.2 and specified  the settings our framework is designed to tackle, in the Limitations.
>
> The revised section now reads:
> ```
> First of all, we mainly investigated the instantiation of our framework based on the MMD, rankings, and
> kernels for rankings. While the type of experimental result is not crucial to our theory, replacing the MMD
> with another statistical distance might invalidate some properties, mainly Theorem 4.2.
> Second, we discuss the key assumptions of the general framework, i.e., of Section 3) and definition 4.1. These
> assumptions ensure that experiments can be run on levels of the generalizability factors selected at random
> within a specific region of C, as it is expected when computing generalizability. First, the experimenter
> should be able to select the levels of all generalizability factors independently from each other. This ensures
> that, if multiple generalizability factors are involved, it is possible to run experiments on the grid of their
> levels. Second, as discussed in Section 4, to compute generalizability in practice the experimenter is supposed
> to run experiments on randomly selected levels of the generalizability factors. If the factors involved are
> unbounded, such a range must be specified in advance by the experimenter. Any claim of generalizability
> outside of this range would follow into the scope of transportabiity.
> Other limitations are specific to the MMD-based instantiation, concerning in particular the validity of Theo-
> rem 4.2. First, the proof of Theorem 4.2 requires that the experimental results follow the same distribution.
> This, in turn, requires that the experimental conditions cannot be chosen sequentially, but have to be sam-
> pled independently from one another. Theorem 4.2 and the analysis given in Section 5 are therefore likely
> not to work if the experiments are performed sequentially, for instance in Active Leaning scenarios. Sec-
> ond, from a more practical perspective, the genexpy module in its current implementation can handle only
> numerical results and rankings. Other data types and kernels require ad-hoc implementations. Third, as
> discussed in Section 5, the analysis might become unfeasible if the number of unique results observed is
> too large. This problem can be lessened by “reducing” the set of results, for instance by discretizing it or
> introducing a minimum performance difference — conceptually similar to the “rope” parameter in Benavoli
> et al. (2017). Fourth and on a similar vein, the proof of Theorem 4.2 (1) requires P to be discrete and
> (2) only holds on the upper tail of the distribution of MMDn. The first assumption is necessary to avoid
> additional approximations in the first step of the proof (see Appendix C.1.2) and does not play a crucial role
> in practice, as the observed support of P is necessarily finite. The second assumption is justified by the fact
> that experimenters are likely to be more interested in achieving higher levels of generalizability.
> To summarize, our framework is designed and suitable to work with a non-sequential factorial experimental
> design, where the experimenter can run experiments on any combination of levels of generalizability factors,
> and where the number of generalizability factors is not too large.
> ```
>
> ## Axis-labeling of Figures 3 and 4
> We apologize for the lack of explicit axes labels.
> We settled on this mainly for two reasons.
> First, the symbols used ($\alpha^*, \delta^*, n^*$) are defined and used multiple times in the sections leading to the figures and each play a distinctive role. Not having these symbols explicitly in the figures might lead to more confusion.
> Second, we experimented with adding both the symbol and the text description to the figure and we found the version with just the symbol to be more esthetically pleasing.
> In the light of the above, we decided to keep only the symbols on the axes labels, while describing them and their role in the captions.

---

### Review · Reviewer_DEUd · 2026-03-24

**Summary Of Contributions:**

The paper contributes a (flexible, extensible) formal framework for analyzing generalizability of the kind of experimental results typically reported in ML papers, including supporting theory, example applications, and an open source package.

Strengths:
- relatively intuitive formalization of a major part of research that's often done informally
- closed form approximations via MMD, and flexible to accommodate different metrics by defining different research-question-appropriate kernels
- concrete examples
- open source code

Weaknesses:
- missing some more explicit discussion of necessary assumptions
- examples are a bit too particular, both including discrete datasets

**Audience:**

Yes

**Audience Explanation:**

The results in this paper have important implications for interpreting, evaluating, and conducting experiments, which is an essential part of much of the TMLR audience's research.

**Claims And Evidence:**

No

**Claims Explanation:**

There's a couple of claims that need a bit more support or explanation---see my requested changes.

**Requested Changes:**

Critical:
- The provided examples make sense, but I'm having a hard time imagining how the framework would be applied to the kind of experiments I most often encounter in my area of ML. In particular, we often simulate datasets using some parametric model, say containing 100 real-valued parameters. We (somewhat arbitrarily) often only consider parameter values in the interval [0.5, 2.0]. Merely talking about the needed "number of experiments" seems inadequate here. Don't we need some assumptions that the experiments are sufficiently broad and sampled appropriately from the distribution of all possible experiments? Where is the discussion and statement of this kind of assumption in the paper? Can this be made more explicit in the Abstract, Section 4, and Section 6?
- On the bottom of Page 6, "Thus, we conclude that the study is more generalizable w.r.t. the Mallows kernel
than the Jaccard kernel": Is comparing the raw kernel values like this really justified? It seems to me you'd rather have to compare them in terms of p-values or some other way of quantifying how the obtained values relate to the underlying distribution of possible values.

Minor:
- The whole approach reminds me of PAC learning. Maybe it's worth adding a reference and offering a bit of discussion about this? Is there some connection between Definitions 4.1/2 and the PAC learnability of the set $\mathcal{X}$?
- The end of Section 2 argues that related work in the causal inference literature is "not applicable to our use-case of ML experimental studies, for which there are arguably neither treatment nor response variables", but then the experimental factors in Section 3.1 are discussed in a way that suggests exactly a causal interpretation of treatment and response variables:
    - "An experimental factor is anything that may affect the result of an experiment"
    - "Design factors are expected to significantly impact the results"
- What kind of run time did the examples in this paper take? Is run time or scalability ever likely to be a concern?
- I see $\mathbb{P}$ defined in Definition 3., but I don't see $\mathbb{P}^n$ defined---it just shows up in Definition 4.1
- Not sure I understand the formula for $n_d$ in the Mallows kernel section. How should I read $| \mathrm{sign}(\cdot) - \mathrm{sign}(\cdot)|$? Does $\mathrm{sign}(\cdot)\$ just return $1$ or $-1$? And then $|\cdot|$ means absolute value here?
- What is $\beta$ in Theorem 4.2?
- Add a reference to [this paper about closed-form expressions for MMD](https://arxiv.org/pdf/1901.03227), e.g., in Section C.1
- In Algorithms 1 and 2, why is the **while** loop comment "Until results aren't generalizable"? Should it rather be "While" instead of "Until", or "are" instead of "aren't"? Or am I misunderstanding the alg?
- replace "cf." with "see"; "cf." [usually indicates](https://en.wikipedia.org/wiki/Cf.) two things should be compared because they differ somehow, while "see" is just a way to reference some supporting information
- In Section 5.1, on line $\kappa_b$, should just be "rank" instead of "ranks"
- On Page 11, add missing word "As [the] study evaluates..."; also use "a priori" instead of "a-priori"

---

> ### Author Response · Authors · 2026-03-30
>
> We thank the reviewer for their detailed and constructive review. The related changes are highlighted in blue in the revised manuscript.
>
> ## Critical 1
> We appreciate this opportunity to improve on the clarity and usefulness of our paper.
> In practice, sampling results independently at random from $\P$ corresponds to running experiments on randomly selected levels of the generalizability factors. If a generalizability factor is bounded (categorical or numerical and bounded), sampling a level at random and proceeding with the experiment is enough. If a generalizability factor is unbounded, the experimenter has to specify a bounded range of admissible values to sample from. The computed generalizability is then valid for the region of $\mathcal C$ defined by the values of the generalizability factors being in the specified ranges.
> We added this description to Section 4 and updated the Limitations and the Abstract accordingly.
>
> ## Critical 2
> We apologize for the confusing statement and agree with the reviewer that, in general, kernels are not intercomparable.
> In the extremely simplified setting of the Example, it is however meaningful to do that.
> Our motivations are (1) that the MMD only depends on the values $\kappa(x, y)$ for $x, y \in \mathcal X$ (see Definition 4.4); (2) $\kappa(x, x) =1$ for all kernels considered, (3) the considered probability distribution only has two possible values.
> Thus, generalizability itself (for a fixed sample size $n$) only depends on the value $\kappa(x, y)$ with $x \neq y$, which are the ones we report in the example.
> To avoid confusions and streamline the example, we removed the direct comparison between kernel values.
>
> ## Minor 1
> We thank the reviewer for pointing out this potential connection to our work.
> We added a discussion about PAC learning to the Future Work section.
> Namely, as PAC theory makes broad statements valid for all
> distributions on a given set X (interpreted as space of results), it might yield lower or upper bounds on the generalizability of any distribution on X , according to whether X is “very easy” or “very hard”.
>
> ## Minor 2
> We apologize for the unclear wording.
> The main difference is that the response variable is often assumed to be either numerical or binary in the causal inference literature, while the same assumption cannot be made for ML experiments.
> For instance, the existing causal inference generalizability techniques would not be able to handle a ranking response variable.
> We updated Section 2 to clarify this point.
>
> ## Minor 3
> The main concerns regard the scalability is that the estimation of the MMD requires quadratic run time.
> In our module, we implemented it so that run time is in $\mathcal O(n_u^2 n_{\text{rep}})$, where $n_u$ is the number of unique results observed and $n_{\text{rep}}$ is the number of repetitions used to compute the MMD.
> We combined this with numpy optimizations to obtain a total run time (including all configurations and all 4 kernels) of 7 minutes for the first case study and 10 for the second, averaging at around 2.5 and 1.75
> seconds per configuration-kernel, respectively.
> We are looking into further optimizations, based for instance on Nyström sampling or a more accurate upper tail approximation of the MMD.
> We added these considerations to Section 5 and in the Limitations.
>
> ## Minors 4, 5
> We apologize for the unclear notation.
> $\mathbb P^n$ is the product probability measure of $n$ independent copies of $\P$, $|\cdot|$ is the standard absolute value on $\mathbb R$, and sign maps a real number $x$ into $1$ if $x > 0$, $-1$ if $x< 0$, and $0$ if $x=0$.
> We clarified their meaning where relevant.
>
> ## Minor 6
> The constant $\beta$, whose explicit expression is in Appendix C1.5, is a real number which depends on the kernel $\kappa$, the distribution of results $\P$, and the desired generalizability $\alpha$.
> Especially important is that $\beta$ does not depend on $n$, preserving the power-law relationship.
> We added this brief description underneath Theorem 4.2.
>
> ## Minor 7
> We thank the reviewer for highlighting a relevant piece of related work.
> While closely related, the paper focuses on (1) numerical distributions, (2) one of which is a multivariate normal, and (3) considering the Gaussian kernel.
> Our approximation, instead, relies on the hypotheses of Theorem 4.2, i.e., (1) that the distributions are discrete and (2) we are in the upper tail of the distribution.
> We added a brief description to Appendix C.1.
>
> ## Minors 8--11
> We thank the reviewer for pointing out these typos and inconsistencies. We have fixed them in the revised manuscript.

---

### Decision · Action_Editor_yaSG · 2026-06-02

**Recommendation:** Reject

**Audience:**

Yes

**Audience Explanation:**

As discussed above, I think it is clear that the paper is in scope for TMLR and would likely be of interest to TMLR's audience.

**Claims And Evidence:**

No

**Claims Explanation:**

This paper proposes a framework for quantifying the generalisability of experimental studies in ML. This is based on treating experimental results as samples from some underlying distribution dictated by the valid experimental conditions and experimental factors.  They then consider the probability of the MMD between results falling below some threshold, thereby indicating that independent experimental runs yield the same result.  They then further consider how large the experimental study needs to be to yield a desired criteria.

The reviews were mixed with two reviewers leaning acceptance and one reviewer quite strongly arguing for rejection.  Overall there was consesus that the problem itself was important, the formalisation a potentially useful way of encapsulating this, the case studies were valuable, and it was beneficial that a code package was provided.  However, there were also concerns with the bold claims made by the paper relative to the concrete contributions made, the assumptions needed for the framework to apply, the limited range of examples, the need for a clearer discussion of limitations, the discussion of related literature and the novelty of the core idea which is quite closely linked to classical frequentist testing and experimental design ideas (though it should be noted that novelty is not a primary criterion for acceptance at TMLR and as the paper has new ideas that may be of interest to the TMLR audience, I believe the paper clearly meets the threshold in this regard).  There was also a concern about the paper's very notion of "generalizability" itself, which is narrower than usually considered in the literature (particularly in ML) and is arguably more about reproducibility instead.

While a number of these concerns were addressed in the revision and rebuttals, I believe that there are some quite serious things still outstanding.  First, I think the paper is quite badly overclaiming at the moment, especially the quite unreasonable suggestion that it "is the first methodology to quantify the generalizability of an experimental study tailored to the ML setting".  While this is easily corrected for, it is a claim the authors have chosen to stick to in the update despite valid objections from reviewers and I think it would be a major problem for it to be published with this claim.  Second, I agree that the paper is not really looking at generalizability in the way that it is thought about in the ML community.  Again, this is in some ways just a question of language, but I think it also hints at an issue in the unsubstantiated novelty claims made by the paper, with huge amounts of work already covering this in the literature (that is, there is a lot of work already discussing reproducibility and I do not think the claim that the paper is taking a substantially different angle to such work is true).  Related to the above, I do not think that the paper adequately positions itself relative to existing literature (such as various works by Demsar, Corani, and or Benavoli, but also more widely the discussion of reproducibility and experimental validation in ML, and more classical statistical testing) and properly explains the distinct contributions relative to this existing work.  Finally, I think that the experimental results in the paper themselves lack context: the case studies demonstrate application of the proposed framework, but do not provide a strong validation of the framework itself compared to possible alternatives. In particular, the paper does not compare against existing statistical-comparison or benchmark-sensitivity methods, nor does it evaluate whether its estimated generalizability predicts stability on held-out experimental conditions.

In light of the issues above, I am inclined to side with the reject reviewer and do not feel the paper is ready to be published in its current form.  While I do think it has genuine positives and see this as quite a close decision, I think the current overclaiming and failure to properly position the contributions in light of existing work mean that a major revision is still required.  I do think the underlying idea is publishable though if these issues can be addressed.

**Resubmission Of Major Revision:**

The authors may consider submitting a major revision at a later time.